# ADAPPROX: MEMORY EFFICIENT OPTIMIZATION VIA ADAPTIVE RANDOMIZED LOW-RANK APPROXIMATION

## ABSTRACT

As deep learning models expand, adaptive learning rate algorithms such as Adam face significant memory consumption challenges due to the need to store of optimizer states, including first and second moment data. Existing memory-efficient methods such as Adafactor and CAME often compromise approximation accuracy with their constant rank-1 matrix factorization techniques. In response, we introduce Adapprox, a novel optimizer that employs adaptive randomized low-rank matrix approximation to more effectively and accurately approximate the second moment. This method dynamically adjusts the rank used for approximation across iterations and weight matrices, mitigating the increase in computation burden while maintaining comparable accuracy. In experiments with GPT-2 and BERT, Adapprox achieves substantial memory savings compared to AdamW and surpasses other memory-efficient counterparts in convergence iterations and downstream task performance, with only a modest increase in the overall latency.

## 1 INTRODUCTION

The emergence of large language models (LLMs) presents substantial memory consumption challenges (Steiner et al., 2023; Shazeer & Stern, 2018; Luo et al., 2023). For instance, BERT (Devlin et al., 2018) consists of up to 300 million parameters, while GPT-3 (Brown et al., 2020) includes as many as 175 billion parameters. Among the most popular optimization algorithms for pretraining LLMs, Adam (Kingma & Ba, 2014) and its variant, AdamW (Loshchilov & Hutter, 2018), intensify these challenges. These optimizers require additional memory to store both first and second moments for each parameter, a feature that contributes to their faster convergence speeds compared to SGD.

To address these challenges, significant efforts have been made to develop memory-efficient optimizers that aim to preserve the advantages of adaptive learning rates while minimizing the memory footprint of optimizer states (Shazeer & Stern, 2018; Anil et al., 2019; Li et al., 2023; Luo et al., 2023). Given the insight that second moment matrices exhibit low-rank characteristics, as shown in Figure 1, applying low-rank matrix approximation techniques can significantly reduce memory footprint during LLM training. Adafactor (Shazeer & Stern, 2018) reduces memory usage by offering an option to omit the first moment and employing a constant rank-1 matrix approximation technique (Anil et al., 2019; Luo et al., 2023) to compress the second moment. However, this approach often results in diminished training effectiveness due to approximation errors. To mitigate these shortcomings, CAME (Luo et al., 2023) builds on Adafactor by introducing a confidence-based scaling factor that matches the dimensions of the second moment. This factor modulates the update step size, slowing it when confidence is low and accelerating it when confidence is high. However, CAME still employs the same factorization method to store confidence statistics to reduce the additional memory consumption, thereby inheriting the fundamental challenges of Adafactor.

Therefore, a key focus is to enhance the method of approximating the second moment to address these ongoing challenges. Figure 1 highlights the shortcomings of the fixed rank-1 approximation. Specifically, the presence of multiple dominant singular values indicates that a rank-1 approximation cannot capture the full complexity of the matrix, leading to reduced accuracy. Although some methods in the literature employ singular value decomposition (SVD) for low-rank approximation by truncating the smallest singular values (Jha & Yadava, 2010; Tsybakov et al., 2011; Kishore Kumar & Schneider, 2017), this technique can be prohibitively time-consuming when iteratively applied to a large number of high-dimensional matrices during LLM training. To overcome the computational

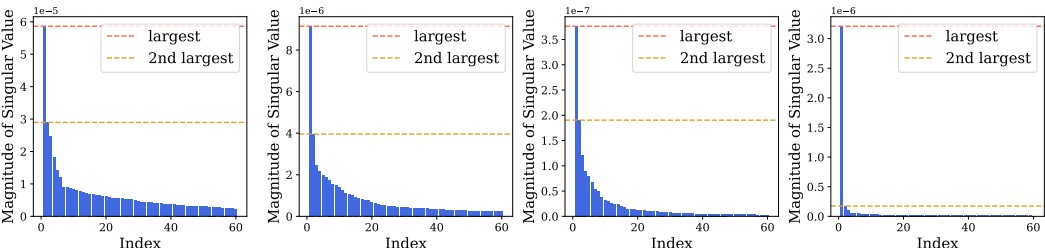

Figure 1: Distribution of singular values in selected second moment matrices. This figure displays the top 60 singular values from 4 second moment matrices, each with a full rank of 1,024, derived from training a GPT-2 345M model using AdamW at the 45,000th iteration.

expense, we employ a Gaussian sampling variant of the randomized SVD algorithm (Halko et al., 2011; Liberty et al., 2007; Li et al., 2014; Batselier et al., 2018), which allows for an efficient low-rank approximation. Additionally, we enhance the approximation accuracy by incorporating oversampling and subspace iteration techniques. Our method enhances computational efficiency compared to the traditional SVD method while achieving approximation accuracy that is on par with it.

Leveraging the principles of randomized low-rank approximation, we introduce Adapprox (Adaptive approximation), a novel method that dynamically adjust the rank for low-rank approximation across iterations and weight matrices, enabling more accurate capture of second moment matrices' features. Specifically, Adapprox assesses approximation accuracy by computing the error ratio between the Euclidean distances of the second moments, which are updated with the true squared gradients and their approximated counterparts. Beginning with an initial rank setting based on memory constraints and an upper bound, this adaptive process ensures that if the error ratio exceeds a predefined threshold, the rank is increased in the next training step to improve accuracy. Conversely, if the error ratio stays below the threshold, the rank is reduced, optimizing computational efficiency and memory usage without sacrificing precision. Additionally, to ensure stable rank adaptation, we introduce a rank moment that employs an exponential averaging mechanism across iterations to determine the new rank. Our experimental results show that our method maintains the initial rank scale during the early stages of training, with the required rank dramatically decreasing in subsequent stages, thereby mitigating the overall training time.

We conduct extensive experiments on pretraining GPT-2 and BERT, as well as a variety of downstream tasks, to demonstrate the effectiveness of Adapprox. Our results show that Adapprox significantly reduces memory usage compared to AdamW. Additionally, it achieves superior convergence iterations, lower validation loss, and enhanced performance in downstream tasks compared to the established state-of-the-art methods, Adafactor and CAME. Moreover, we observed only a modest increase in latency with Adapprox due to the introduced randomized low-rank approximation steps, which is a small price to pay for the benefits gained. All of the above supports Adapprox's overall superiority in terms of both efficiency and effectiveness.

## 2 RELATED WORK

**Low-Rank Matrix Approximation.** Low-rank matrix approximation aims to represent a matrix using lower-rank matrices, seeking more efficient data representation while retaining as much information as possible. For example, a rank-$k$ approximation can represent an $m \times n$ matrix while reducing the memory footprint from $\mathcal{O}(mn)$ to $\mathcal{O}(k(m + n))$. This technique is utilized across a wide spectrum of applications, including principal component analysis (Shen & Huang, 2008; Papailiopoulos et al., 2013), image processing (Haeffele et al., 2014; Guo et al., 2017; Chen et al., 2017), and various machine learning scenarios (Paterek, 2007; Li et al., 2016).

**Memory Efficient Optimizers.** Memory-efficient optimizers aim to reduce memory usage by compressing optimizer states during training, ideally without affecting performance. To achieve this goal, various strategies have been proposed, including low-rank matrix approximation (Shazeer & Stern, 2018; Luo et al., 2023) and quantization techniques (Li et al., 2023). Notably, these two

categories of methods are orthogonal and can be integrated seamlessly. Adafactor (Shazeer & Stern, 2018) and CAME (Luo et al., 2023) are notable examples that utilize low-rank matrix approximation, significantly reducing memory usage, though this can come at the expense of accuracy due to their rank-1 factorization approaches. Quantization techniques, as employed in 8-bit Adam (Dettmers et al., 2021) and 4-bit Adam (Li et al., 2023), offer another avenue for memory savings by reducing the precision of stored values without drastically affecting performance. In this paper, we focus on enhancing low-rank matrix approximation techniques to reduce memory usage while maintaining high accuracy and performance during LLM training.

## 3 METHODOLOGY

This section begins with an overview of Adam and AdamW to contextualize the necessity for compressing the optimizer state when training LLMs. We then introduce two key components of our method: the randomized low-rank approximation for the second moment and the adaptive rank selection mechanism. Finally, we provide a detailed description of the Adapprox optimizer.

### 3.1 OVERVIEW OF THE ADAM AND ADAMW

Consider a function $f(W)$, where $W \in \mathbb{R}^{m \times n}$ denotes the parameters of the neural network. The update rule for Adam (Kingma & Ba, 2014) at the $t$-th iteration is defined as follows:

$$(\text{Adam}) \begin{cases} G_t = \nabla f\left(W_{t-1}\right), & \text{(1a)} \\ M_t = \beta_1 M_{t-1} + (1 - \beta_1)G_t, & \text{(1b)} \\ V_t = \beta_2 V_{t-1} + (1 - \beta_2)G_t^2, & \text{(1c)} \\ \widehat{M_t} = M_t / \left(1 - \beta_1^t\right), & \text{(1d)} \\ \widehat{V_t} = V_t / \left(1 - \beta_2^t\right), & \text{(1e)} \\ W_t = W_{t-1} - \alpha \widehat{M_t} / \left(\sqrt{\widehat{V_t}} + \epsilon\right). & \text{(1f)} \end{cases}$$

Here, all computations are element-wise. $G_t$ represents the gradient arranged in matrix form. $M_t$ and $V_t$ are the exponential running averages of the first and second moments, respectively. $\widehat{M_t}$ and $\widehat{V_t}$ are the bias-corrected versions of $M_t$ and $V_t$. The parameters $\beta_1$ and $\beta_2$ control these moment estimates. Additionally, $\alpha$ denotes the learning rate, and $\epsilon$ is a small positive constant introduced to prevent division by zero. Building upon Adam, AdamW (Loshchilov & Hutter, 2018) decouples weight decay from the gradient updates. With this change, the parameter update step in AdamW is

$$W_t = W_{t-1} - \alpha \left(\widehat{M_t} / \left(\sqrt{\widehat{V_t}} + \epsilon\right) + \lambda W_{t-1}\right), \tag{2}$$

where $\lambda$ is the rate of the weight decay. Adam and AdamW require the storage of both $M_t$ and $V_t$ at each step, which necessitates $\mathcal{O}(mn)$ extra memory compared with SGD.

### 3.2 RANDOMIZED LOW-RANK APPROXIMATION FOR THE SECOND MOMENT

Adafactor (Shazeer & Stern, 2018) provides the option to eliminate the first moment entirely, which may lead to slower convergence rates (see Appendix A). In contrast, the second moment typically exhibits low-rank characteristics, as illustrated in Figure 1. This property enables the application of efficient compression techniques, such as low-rank approximation. Therefore, we concentrate on compressing the second moment matrices to reduce the memory footprint required for storage.

For a matrix $A \in \mathbb{R}^{m \times n}$, deriving its low-rank approximation can be formulated as an optimization problem:

$$\min_{Q,U} \|A - QU^\top\|_F^2, \tag{3}$$

where $\|\cdot\|_F$ is the Frobenius norm and $Q \in \mathbb{R}^{m \times k}$ and $U \in \mathbb{R}^{n \times k}$ are two feature matrices with $1 \leq k \ll \min\{m, n\}$. Then, the resulting matrix $A_k = QU^\top$ servers as the rank-$k$ approximation

---

**Algorithm 1** Streamlined Randomized Subspace Iteration

---

**Inputs:** Target matrix $A \in \mathbb{R}^{m \times n}$, target rank $k$
$U \sim \mathcal{N}(0,1)^{n \times (k+p)}$          # generate $U$ from a standard Gaussian distribution
$Q, R \leftarrow \mathbf{0}^{m \times (k+p)}, \mathbf{0}^{(k+p) \times (k+p)}$          # initialize $Q$ and $R$
**for** $i \leftarrow 1, 2, \ldots, l$ **do**
     $Q \leftarrow AU$          # compute $Q$ as a random sample from the column space of $A$
     $Q, R \leftarrow$ QR decomposition$(Q)$          # transform $Q$ into an orthogonal matrix
     $U \leftarrow A^\top Q$          # compute $U$ as the projection of $Q$ onto the row space of $A$
**end for**
**return** $Q[:, :k], U[:, :k]$

---

of $A$. The optimal $A_k$ can be determined through a complete SVD of $A$, followed by truncation to retain only the top $k$ singular values and their corresponding singular vectors. This yields the representation (Golub & Van Loan, 2013):

$$A_k = \sum_{i=1}^{k} \sigma_i u_i v_i^T, \quad \|A - A_k\|_F^2 = \sum_{i=k+1}^{\min\{m,n\}} \sigma_i^2, \tag{4}$$

where $\sigma_1 \geq \sigma_2 \geq \cdots \geq \sigma_{\min\{m,n\}} \geq 0$ are singular values of $A$, and $u_i$ and $v_i$ are corresponding left and right singular vectors.

Nevertheless, computing the full SVD for large matrices poses significant computational and memory challenges (Jha & Yadava, 2010; Tsybakov et al., 2011; Kishore Kumar & Schneider, 2017). To address these issues, we employ randomized low-rank matrix approximation algorithms (Liberty et al., 2007; Halko et al., 2011; Nakatsukasa, 2020), which provide a balance between computational efficiency and memory economy while yielding high-quality low-rank approximations. Specifically, our implementation utilizes the Gaussian sampling variant of the randomized SVD algorithm (Halko et al., 2011). In this approach, we bypass singular value estimation to streamline the process, focusing solely on the extraction of feature matrices. The comprehensive procedure of this modified method is detailed in Algorithm 1, termed Streamlined Randomized Subspace Iteration (S-RSI).

The objective of S-RSI is to compute an approximate basis $Q \in \mathbb{R}^{m \times k}$ with orthonormal columns for the column space of the target matrix $A \in \mathbb{R}^{m \times n}$, such that

$$A_k = QQ^\top A. \tag{5}$$

Accoring to Equation 3, we then form $U = Q^\top A$, resulting in $Q \in \mathbb{R}^{m \times k}$ and $U \in \mathbb{R}^{n \times k}$. To achieve this, we compute $Q$ efficiently using random sampling methods. Consider drawing a random vector $u$, where each element is independently and identically distributed according to a standard Gaussian distribution. Then, the computation $q = Au$ serves as a stochastic representation of the column space of $A$, as $q$ represents a random linear combination of the columns of $A$. By repeating this sampling process $k$ times, we obtain a set of random vectors:

$$\{q_i \mid q_i = Au_i, \ i = 1, 2, \ldots, k\}. \tag{6}$$

Due to the inherent randomness in the generation of $u_i$, the set of vectors $\{u_i\}_{i=1}^{k}$ is expected to occupy a general linear position, which implies a high likelihood that any subset of these vectors is linearly independent. Consequently, this observation leads us to propose the following:

**Proposition 3.1.** *Given a set of randomly generated vectors $\{u_i\}_{i=1}^{k}$ that are in a general linear position, and a full-rank matrix $A \in \mathbb{R}^{m \times n}$, the set of vectors $\{q_i \mid q_i = Au_i\}_{i=1}^{k}$ are also linearly independent.*

*Proof.* Linear independence of $\{u_i\}_{i=1}^{k}$ implies that $\sum_{i=1}^{k} a_i u_i = 0$ holds only when all scalars $\{a_i\}_{i=1}^{k}$ are zero. We examine a linear combination of the vectors $\{q_i\}_{i=1}^{k}$:

$$\sum_{i=1}^{k} a_i q_i = \sum_{i=1}^{k} a_i Au_i = A \sum_{i=1}^{k} a_i u_i. \tag{7}$$

Since $A$ is full rank, $\sum_{i=1}^{k} a_i q_i \neq 0$ unless all $a_i$ are zero, indicating that vectors $\{q_i\}_{i=1}^{k}$ are linearly independent. □

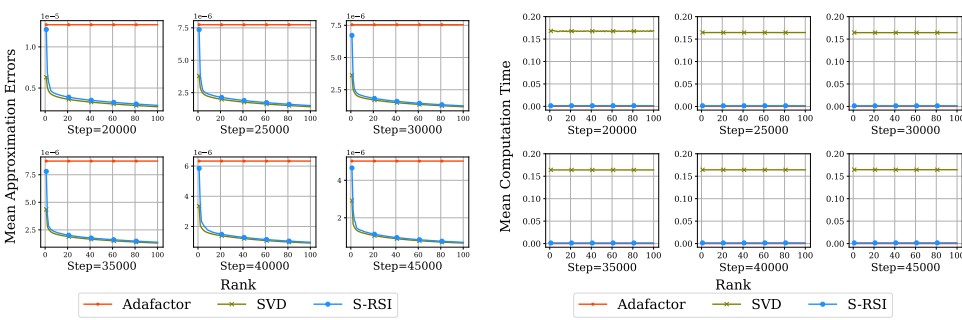

(a) Mean Approximation Error vs. Rank $k$.  (b) Mean Computation Time vs. Rank $k$.

Figure 2: Comparative analysis of the S-RSI ($l = 5$ and $p = 5$) against Adafactor and full SVD. All methods are applied to the second-moment matrices derived from training a GPT-2 345M model using the AdamW, with results captured at various stages of the training process. (a) shows that S-RSI achieves similar approximation performance to full SVD; (b) shows that the time cost of S-RSI (in seconds) is significantly lower than that of full SVD and remains approximately constant as the rank $k$ increases.

By Proposition 3.1, the set $\{q_i\}_{i=1}^k$ can be arranged as the columns of the matrix $Q$, followed by the application of an orthonormalization procedure such as QR decomposition (Golub & Van Loan, 2013). To enhance the robustness of our approximation, we incorporate an oversampling mechanism by sampling $k + p$ columns for $Q$, where $k$ is the target rank and $p$ is a small oversampling parameter (e.g., $p = 5$; see Appendix B for details on the selection of $p$). By sampling $k + p$ columns, we capture a richer set of directions in the original space. This approach mitigates the effects of numerical instability and ensure a more complete representation of the relevant subspace, ultimately leading to improved accuracy in the low-rank approximation.

Power iteration (Rokhlin et al., 2010; Halko et al., 2011; Golub & Van Loan, 2013) is a technique designed to enhance the singular values of a matrix, allowing us to distinguish more effectively between the significant and less significant components. This is particularly beneficial in low-rank approximations, where the goal is to capture the most relevant features of the data while ignoring noise or less informative structures. By applying the power iteration method, we seek to amplify the singular values of the target matrix $A$. The key is to perform multiple applications of the transformation, which increases the prominence of larger singular values. Mathematically, this can be represented as:

$$A^l = (Q\Sigma U^T)^l = Q\Sigma^l U^T, \tag{8}$$

where $l$ is a positive integer that indicates the number of iterations (e.g., $l = 5$; see Appendix B for details on the selection of $l$). The effect is that the singular values $\sigma_i$ in $\Sigma$ are raised to the $l$-th power, emphasizing larger values. In our context, we specifically consider the modified matrix

$$A' = (AA^\top)^l A = (Q\Sigma U^\top U\Sigma Q^\top)^l Q\Sigma U^\top = Q\Sigma^{2l+1}U^\top, \tag{9}$$

where the term $\Sigma^{2l+1}$ indicates that the singular values grow exponentially with respect to $l$, significantly differentiating the larger singular values from the smaller ones. Therefore, by utilizing power iteration, the significant singular values become disproportionately larger compared to less significant ones. This allows for a more robust approximation of the original matrix $A$.

Following the randomized SVD algorithm (Halko et al., 2011), the approximation error bound is:

$$\mathbb{E}\|A - QU^\top\| \leq \left[ \left( 1 + \sqrt{\frac{k}{p-1}} \right)^{2l+1} \sigma_{k+1}^{2l+1} + \frac{e\sqrt{k+p}}{p} \sqrt{\sum_{j>k} \sigma_j^{2(2l+1)}} \right]^{1/(2l+1)}. \tag{10}$$

According to Equation 10, the approximation error can be reduced by increasing $k$, $p$, and $l$. Using the S-RSI method, we can efficiently compress the storage of matrix $A$ from $\mathcal{O}(mn)$ to $\mathcal{O}(k(m+n))$ with a time complexity of $\mathcal{O}(lmn(k+p))$. In contrast, the time complexity for computing the full SVD is $\mathcal{O}(mn^2 + m^2n)$.

| | $m = n = 1024$ | $m = n = 2048$ | $m = n = 4096$ | $m = n = 5120$ | $m = n = 8192$ |
|---|---|---|---|---|---|
| $k = 8$ | $7.6 \times 10^{-4}$s | $7.5 \times 10^{-4}$s | $7.8 \times 10^{-4}$s | $7.8 \times 10^{-4}$s | $8.2 \times 10^{-4}$s |
| $k = 64$ | $8.8 \times 10^{-4}$s | $9.5 \times 10^{-4}$s | $7.7 \times 10^{-4}$s | $7.7 \times 10^{-4}$s | $7.9 \times 10^{-4}$s |
| $k = 256$ | $1.5 \times 10^{-3}$s | $1.4 \times 10^{-3}$s | $2.1 \times 10^{-3}$s | $2.1 \times 10^{-3}$s | $2.0 \times 10^{-3}$s |

Table 1: Actual time cost of S-RSI with various model size and rank settings, where $p = l = 5$.

We demonstrate the efficacy of S-RSI through empirical comparisons with Adafactor and full SVD. Our experiments utilize second-moment matrices obtained during the training of a GPT-2 345M model with AdamW as the target matrices. Figure 2 shows the mean approximation error and computation time across varying ranks. The $x$-axis represents the rank set for S-RSI and full SVD, while Adafactor employs its fixed rank-1 approximation.

Both full SVD and S-RSI show significant error reductions with modest rank increases compared to Adafactor, with S-RSI nearing the performance of full SVD. This observation underscores the limitation of relying on fixed rank-1 approximation and highlights the substantial benefits in approximation accuracy that can be achieved with only a slight increase in rank $k$.

In terms of computation time, Adafactor remains the most efficient, while S-RSI significantly reduces computation time compared to full SVD. Although S-RSI theoretically exhibits a linear relationship with rank $k$ and a quadratic relationship with increases in $m$ and $n$, our observations indicate that the actual time consumption on GPUs is less significant than theoretically expected. We conduct an experiment to evaluate the actual time consumption of S-RSI for compressiong a matrix in relation to $k$, $m$, and $n$ on a NVIDIA A100 GPU with $p = l = 5$. This assessment considers model sizes ranging from GPT-2 345M (hidden size 1024) to LLaMA-3-70B (hidden size 8192), focusing on the additional time required for larger models. As shown in Table 1, the time cost does not follow the theoretically expected relationship with increases in $k$, $m$, and $n$. We attribute this discrepancy to the parallel computation capabilities of the hardware. Overall, S-RSI effectively balances approximation accuracy and computational efficiency.

### 3.3 ADAPTIVE RANK SELECTION

The results in Figure 2 underscore the importance of selecting the appropriate rank $k$ for low-rank approximation of the second moment. Choosing a larger $k$ can result in increased computational demands for marginal precision improvements, while a smaller $k$ risks significantly compromising accuracy. To address this, we develop an adaptive rank selection mechanism that dynamically adjusts $k$ for each target matrix through iterations for S-RSI.

Specifically, we employ a step-wise reflection rank control mechanism. At each step after runing S-RSI, we evaluate the approximation error ratio by

$$\xi = \frac{\|A - QU^\top\|_F}{\|A\|_F}. \tag{11}$$

If $\xi > \xi_{thresh}$, it implies that $k$ is not large enough to meet the approximation precision requirement, and the rank should be increased in the next iteration. Conversely, if $\xi \leq \xi_{thresh}$, it implies that $k$ is sufficient for the approximation precision requirement, and the rank should be decreased in the next iteration. Additionally, we impose a constraint on $k_{next}$ to ensure that $k_{min} \leq k_{next} \leq k_{max}$. The pseudocode for this method is summarized in Algorithm 2, which we designate as Adaptive S-RSI (AS-RSI). In our experiments, we set the step size $\Delta k$ to $\max\{\lceil 0.02 k_{max} \rceil, 1\}$.

### 3.4 ADAPPROX ALGORITHM

The integration of the proposed methodologies culminates in the Adapprox algorithm defined in Algorithm 3. At each step, $f_t$ represents a stochastic realization of the objective function $f$, exemplified by the loss function computed using a randomly selected mini-batch of data. We then compute the gradient $G_t$ relative to the previous parameters and the exponential running averages of the second moment $V_t$. Note that $V_{t-1}$ is reconstructed from the product $Q_{t-1}U_{t-1}^\top$, after which $V_t$ is compressed using the AS-RSI method. Following this, the moment of rank $k$ is computed

---

**Algorithm 2** Adaptive S-RSI

**Inputs:** Target matrix $A \in \mathbb{R}^{m \times n}$, rank $k$, the bound $k_{min}$ and $k_{max}$, and $\xi_{thresh}$
$Q, U^\top \leftarrow$ S-RSI$(A, k)$
$\xi \leftarrow \|A - QU^\top\|_F / \|A\|_F$          # evaluate the error ratio based on the current rank $k$
**if** $\xi > \xi_{thresh}$ **then**
   $k_{next} \leftarrow k + \Delta k$       # increase $k$ if the current error ratio exceeds an acceptable threshold
**else**
   $k_{next} \leftarrow k - \Delta k$           # decrease $k$ if the current error ratio is acceptable
**end if**
$k_{next} \leftarrow \min\{k_{max}, \max\{k_{min}, k_{next}\}\}$
**return** $Q[:, :k], U[:, :k], k_{next}$

---

**Algorithm 3** Adapprox

**Inputs:** Initial point $W_0 \in \mathbb{R}^{m \times n}$, $M_0 = \mathbf{0}^{m \times n}$, $Q_0 = \mathbf{0}^{m \times k_0}$, and $U_0 = \mathbf{0}^{n \times k_0}$, learning rates $\{\alpha_t\}_{t=1}^T$, moment decay $\beta_1$ and $\beta_2$, small constant $\epsilon$, clipping threshold $d$, initial rank $k_0$, the bounds of rank $k_{min}$ and $k_{max}$, integer $l$, integer $p$ with $(k_0 + p) \leq k_{max}$, moment decay $\beta_3$ for rank adaption, threshold $\xi_{thresh}$, and weight decay rate $\lambda$
**for** $t \leftarrow 1, 2, \ldots, T$ **do**
   $G_t \leftarrow \nabla f_t(W_{t-1})$                    # compute the gradient
   $V_t \leftarrow \beta_2 [Q_{t-1} U_{t-1}^\top]_+ + (1 - \beta_2) G_t^2$     # compute the second moment $V_t$
   $Q_t, U_t^\top, k_t \leftarrow$ AS-RSI$(V_t, k_{t-1}, k_{min}, k_{max}, \xi_{thresh})$    # compress $V_t$ for the next iteration
   $k_t \leftarrow \beta_3 k_{t-1} + (1 - \beta_3) k_t$                # update the rank $k$
   $M_t \leftarrow G_t / (\sqrt{V_t} + \epsilon)$
   $M_t \leftarrow M_t / \max(1, \text{RMS}(M_t)/d)$         # the clipping mechanism
   **if** $\beta_1 > 0$ **then**
      $M_t \leftarrow \beta_1 M_{t-1} + (1 - \beta_1) M_t$     # compute the first moment if necessary
   **end if**
   $W_t \leftarrow W_{t-1} - \alpha_t(M_t + \lambda W_{t-1})$         # update the weights
**end for**

---

to smoothly adjust the rank value for approximation, as detailed in Section 3.3. Subsequently, we calculate the update $M_t = G_t / (\sqrt{V_t} + \epsilon)$ and incorporate the update clipping mechanism as proposed in Adafactor (Shazeer & Stern, 2018) to mitigate excessively large updates:

$$M_t \leftarrow \frac{M_t}{\max(1, \text{RMS}(M_t)/d)}, \quad \text{RMS}(M_t) = \frac{\|M_t\|_F}{\sqrt{mn}},$$

where $d$ is the clipping threshold. We also offer the option to omit the first moment, depending on whether $\beta_1$ is set to zero. Finally, parameter updates are executed in a decoupled weight decay fashion, as delineated in Equation 2.

## 4 EXPERIMENTS

We investigate GPT-2 117M/345M (Radford et al., 2019) and BERT 345M models (Devlin et al., 2018). Our pretraining experiments utilize The Pile dataset (Gao et al., 2020) and the SentencePiece tokenizer (Kudo & Richardson, 2018). We evaluate the pretrained models on several downstream tasks, including Arc Easy (Clark et al., 2018), HellaSwag (Zellers et al., 2019), QQP (Quora, 2017), SST-2 (Socher et al., 2013), RTE (Dagan et al., 2006), CoLA (Warstadt et al., 2018), MRPC (Dolan & Brockett, 2005), QNLI (Wang et al., 2018), and WNLI (Wang et al., 2018). Our primary baselines include AdamW (Loshchilov & Hutter, 2018), Adafactor (Shazeer & Stern, 2018), and CAME (Luo et al., 2023). We have implemented our optimization algorithm using the PyTorch framework (Paszke et al., 2019). Additionally, the pretraining of GPT-2 is conducted utilizing the Megatron-LM framework (Shoeybi et al., 2019) and eight NVIDIA Tesla V100 GPUs.

For GPT-2 pretraining, we set $\beta_1$, $\beta_2$, and $\beta_3$ at 0.9, 0.999, and 0.9, respectively, and maintain a consistent weight decay rate of 0.1 for all compared algorithms. Adapprox's additional parameters are specified as follows: $\epsilon = 1 \times 10^{-8}$, $k_0 = 256$, $k_{min} = 1$, $k_{max} = k_0$, and $\xi_{thresh} = 1 \times 10^{-3}$. For Adafactor and CAME, the other parameters are set to their respective default values. We adopt a linear warmup strategy followed by a cosine-style learning rate decay, both integrated within the

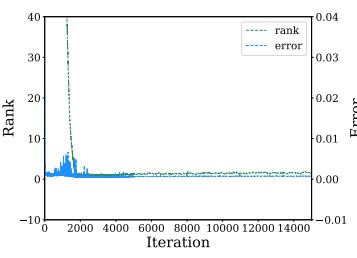
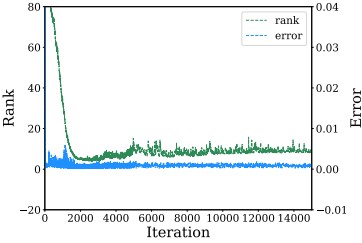

(a) GPT-2 117M, rank vs. iteration.    (b) BERT 345M, rank vs. iteration.

Figure 3: Assessment of the rank adaptation mechanism during the training of GPT-2 117M and BERT 345M. For clarity, we omit the details of the first 1,000 iterations, during which $k$ decreases from $k_0 = k_{max} = 256$.

Megatron-LM framework. To guarantee fair comparisons, all evaluated optimizers use uniform training parameters for each model, selected through empirical testing and established best practices. This also allows us to isolate the impact of algorithmic differences rather than that of hyperparameter tuning, ensuring a consistent basis for comparison. Specifically, the sequence length, batch size, number of training iterations, number of warmup iterations, peak learning rate, and minimum learning rate are set as follows: for GPT-2 117M, they are 1024, 128, 100K, 1K, $3 \times 10^{-4}$, and $5 \times 10^{-5}$; for GPT-2 345M, 1024, 128, 100K, 1K, $3 \times 10^{-4}$, and $3 \times 10^{-5}$; and for BERT 345M, 1024, 128, 200K, 2K, $2 \times 10^{-4}$, and $2 \times 10^{-5}$, respectively.

For downstream tasks, we individually adjust the learning rates within the range $[2 \times 10^{-5}, 4 \times 10^{-5}, 5 \times 10^{-5}]$ for GPT-2 117M and $[1 \times 10^{-5}, 5 \times 10^{-5}, 1 \times 10^{-4}, 2 \times 10^{-4}]$ for BERT 345M, fine-tuning models that were pretrained with each evaluated optimizer. We then select the best results achieved under these learning rates. In addition, we conduct 3 fine-tuning epochs for the GPT-2 117M and 1 epoch for the BERT 345M models. The bacth size is set to 128, the sequence length is set to 1024, the learning rate scheduler follows a cosine decay style, and the warmup ratio is set to 0.01.

### 4.1 ASSESSMENT OF RANK ADAPTATION

We investigate the variation in rank by AS-RSI during the training of GPT-2 117M and BERT 345M, with the results visualized in Figure 3. For clarity, we omit the details of the first 1,000 iterations, during which $k$ decreases from $k_0 = k_{max} = 256$. In the subsequent iterations, the rank $k$ fluctuates around a stable level. Furthermore, the mean rank during the training of GPT-2 117M is approximately 1.4, whereas for BERT 345M, it is 7.7. This finding shows substantial memory savings of Adapprox compared to the AdamW optimizer, which utilizes the full rank of the second moment at 1024. We report the peak memory usage of Adapprox and memory usage during the most of later iterations in Table 2.

|  | GPT-2 117M | | BERT 345M | |
|---|---|---|---|---|
| Method | $k = 256$ | $k = 2$ | $k = 256$ | $k = 8$ |
| AdamW | 819.18 MB | | 2565.75 MB | |
| Adapprox | **583.93 MB** | **412.77 MB** | **1704.62 MB** | **1297.39 MB** |

Table 2: Peak memory usage of Adapprox, along with memory usage during the majority of later iterations for training GPT-2 117M and BERT 345M.

As illustrated above, the peak memory usage is related to $k_{max}$ (also $k_0$). In practice, the selection of this upper bound may depend on specific models and available resources. Assuming $m = tn$, the theoretical memory saving ratio is $\mathcal{O}(k(t + 1)/tn)$. In practice, we ensure that $k$ is set smaller than $n$ to guarantee memory savings. We provide an illustration of optimizer state memory usage with compressed second moments across various values of $k$ and model sizes in Appendix C.1.

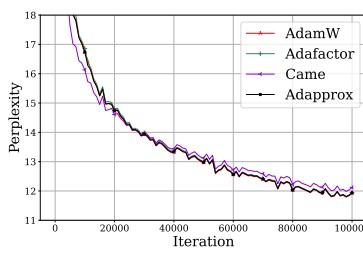
(a) GPT-2 117M, perplexity vs. iter.

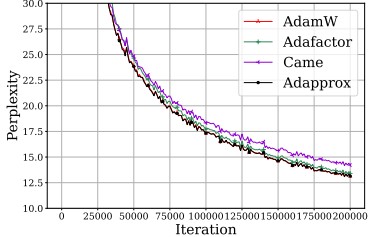
(b) BERT 345M, perplexity vs. iter.

Figure 4: Comparative analysis of Adapprox against AdamW, Adafactor, and CAME for pretraining.

| Model | Method | Arc-e | HellaSwag | QQP | SST-2 | Average |
|-------|--------|-------|-----------|-----|-------|---------|
| GPT-2 117M | AdamW | 24.96 | 26.62 | **63.18** | 50.46 | 41.30 |
| | Adafactor | 25.00 | 26.74 | 63.18 | 52.41 | 41.83 |
| | CAME | 25.59 | 26.67 | 63.17 | 50.92 | 41.59 |
| | Adapprox | **26.05** | **26.79** | 63.16 | **53.56** | **42.39** |
| GPT-2 345M | AdamW | 25.38 | 26.70 | **63.18** | 50.92 | 41.54 |
| | Adafactor | **26.52** | 26.52 | 63.16 | 51.61 | 41.95 |
| | CAME | 24.96 | 26.51 | **63.18** | 46.67 | 40.33 |
| | Adapprox | 26.05 | **26.79** | 63.16 | **53.56** | **42.39** |

Table 3: Zero-shot results(accuracy, ↑) of pretrained GPT-2 models with various optimizers.

| | AdamW | Adafactor | CAME | Adapprox |
|--------|-------|-----------|------|----------|
| per iteration | 1.81s | 1.84s | 1.85s | 1.92s |
| 10K iterations | 5.10h | 5.23h | 5.27h | 5.41h |

Table 4: Time costs of methods applied to GPT-2 117M ("s" stands for seconds and "h" for hours).

## 4.2 GPT-2 AND BERT TRAINING

We present the convergence curves for validation loss and perplexity on GPT-2 117M and BERT 345M, zero-shot results on two GPT-2 models, and the associated time cost to demonstrate the effectiveness of Adapprox for pretraining.

Figure 4 compares the performance of Adapprox with AdamW, Adafactor, and CAME during the pretraining of GPT-2 117M and BERT 345M models, illustrating the validation perplexity. Compared to AdamW, Adapprox generally exhibits comparable results. Although CAME initially demonstrates lower perplexity, it tends to converge to suboptimal results as training progresses. These findings suggest that Adapprox effectively balances accuracy with memory usage and may provide faster convergence and superior performance relative to Adafactor and CAME.

The results of zero-shot tasks on pretrained GPT models are detailed in Table 3. These results indicate that Adapprox surpasses existing methods on most tasks. Furthermore, when evaluating the average accuracy across the four tasks, Adapprox consistently shows superior performance compared to existing methods. The superior performance of Adapprox and Adafactor compared to AdamW in zero-shot scenarios can be attributed to the clipping mechanism implemented in both, as discussed in Section 3.4. This mechanism effectively address outdated second moment estimators in AdamW.

In Table 4, we present a comparison of running time per iteration and across 10K iterations (with evaluations conducted every 1,000 iterations) for the methods applied to GPT-2 117M. There is only a modest latency increase of approximately 6% with Adapprox compared to Adam, which is a small trade-off for the memory savings achieved.

| Model | Method | RTE | CoLA | MRPC | QNLI | SST-2 | WNLI | Average |
|-------|--------|-----|------|------|------|-------|------|---------|
| GPT-2 117M | AdamW | 64.62 | 76.61 | 76.61 | 87.31 | 89.91 | **56.34** | 75.23 |
| | Adafactor | 65.34 | 75.46 | 80.64 | 87.17 | 89.91 | 50.70 | 74.87 |
| | CAME | 57.76 | 69.13 | 76.96 | 64.10 | 81.77 | **56.34** | 67.68 |
| | Adapprox | **68.95** | **80.63** | **84.31** | **88.50** | **91.51** | **56.34** | **78.38** |
| BERT 345M | AdamW | **62.82** | **77.26** | **78.19** | 89.90 | 91.06 | 53.52 | **75.45** |
| | Adafactor | 54.51 | 73.70 | 75.49 | 89.79 | 90.37 | 46.47 | 71.72 |
| | CAME | 58.85 | 71.88 | 75.49 | 86.91 | **91.63** | **56.34** | 73.52 |
| | Adapprox | 62.46 | 74.38 | 76.96 | **89.91** | 90.25 | **56.34** | 75.05 |

Table 5: Fine-tuning performance (accuracy, ↑) of the compared optimizers for GPT-2 117M and BERT 345M models across various downstream tasks.

## 4.3 DOWNSTREAM TASKS

We evaluate the downstream task performance of GPT-2 117M and BERT 345M models, each pretrained and fine-tuned with its corresponding optimizer. The empirical results are presented in Table 5. Our experimental findings underscore the superiority of Adapprox compared to those using Adafactor and CAME. Besides, Adapprox not only achieves performance comparable to AdamW but also surpasses it in certain cases.

## 5 DISCUSSION

While Adapprox and Adafactor achieve significant memory savings through the low-rank approximation of the second moment and by omitting the first moment, our experiments demonstrate that the absence of the first moment has a notable impact on convergence speed and overall performance (see Appendix A). Consequently, we recommend retaining the first moment unless memory constraints are extremely prohibitive. In the future, we aim to further reduce the time cost of Adapprox by decreasing the frequency of rank adjustments in the later stages of training, where we observe that the rank remains nearly constant (as shown in Figure 3).

## 6 CONCLUSION

In this paper, we present Adapprox, a novel optimizer engineered to mitigate memory consumption challenges inherent in training large-scale models. We employ S-RSI, which leverages randomized low-rank matrix approximation to reduce the memory footprint of the second moment. We observe that the time cost of S-RSI is less significant in practice than theoretically expected, attributed to the parallel computation capabilities of GPUs. Furthermore, we enhance S-RSI with an adaptive rank selection mechanism, introducing AS-RSI that dynamically adjusts the rank value throughout the training iterations. These components are integrated to formulate Adapprox. Our empirical evaluations, encompassing both pretraining and downstream tasks for GPT-2 117M/345M and BERT 345M models, corroborate the efficacy of Adapprox. In comparison to AdamW, Adapprox delivers substantial memory savings via its low-rank approximation strategy. Although there is a minor trade-off in memory efficiency compared to state-of-the-art memory efficient optimizers such as Adafactor and CAME, Adapprox surpasses these competitors in several critical performance metrics, including validation loss and perplexity during pretraining, along with accuracy in downstream tasks. These results position Adapprox as a balanced solution for memory-efficient training, successfully balancing efficiency with minimal accuracy sacrifices.

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

## A    ANALYSIS OF FIRST MOMENT EFFICACY

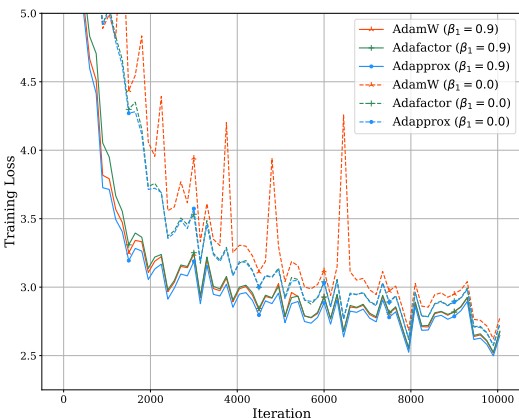

Figure 5: Training loss vs. iteration for AdamW, Adafactor, and Adapprox optimizers, comparing scenarios with and without the first moment.

Figure 5 demonstrates that incorporating the first moment significantly accelerates the convergence process, evidenced by achieving lower training losses at the same iteration, for each optimizer examined, including AdamW, Adafactor, and Adapprox. CAME is omitted from this analysis due to its incompatibility with $\beta_1 = 0$. Furthermore, while AdamW exhibits instability without the first moment, Adafactor and Adapprox mitigate this through the use of a clipping mechanism, effectively reducing large, unexpected updates and enhancing stability.

## B    ANALYSIS OF PARAMETERS IN S-RSI AND AS-RSI

The parameters $k_0$, $k_{\min}$, $k_{\max}$, $l$, and $p$ in S-RSI and AS-RSI are chosen to balance approximation accuracy, memory efficiency, and computational cost.

Regarding approximation accuracy, Equation 10 demonstrates the impact of these parameters on the low-rank approximation. Larger values of $k$, $l$, and $p$ typically lead to smaller approximation errors, as shown in Equation 10, but also increase computational cost and memory usage.

First, $l$ denotes the number of power iterations in the S-RSI method. Increasing $l$ enhances approximation accuracy but also raises computational cost due to the sequential nature of power iterations. Our experiments indicate that $l = 5$ strikes a good balance between accuracy and efficiency. The table below presents the results of an ablation study on $l$, where $A$ is an $m \times n$ matrix sampled from a normal distribution:

Table 6: Analysis on $l$, where $A$ is an $m \times n$ matrix sampled from a normal distribution ($m = 1024$, $n = 1024$, $k = 256$, $p = 5$).

| $l$ | 1 | 2 | 3 | 4 | 5 | 6 | 7 | 8 | 9 | 10 |
|---|---|---|---|---|---|---|---|---|---|---|
| $\frac{\|A - QU^\top\|_F}{\|A\|_F}$ | 0.1473 | 0.0968 | 0.0850 | 0.0828 | 0.0802 | 0.0786 | 0.0783 | 0.0779 | 0.0779 | 0.0778 |

As shown, $l = 5$ provides sufficient accuracy, with further increases offering only marginal gains. Thus, we recommend $l = 5$ in our configuration. Similarly, we recommend setting $p = 5$ in our configuration:

Additionally, $k_0$, $k_{\max}$, and $k_{\min}$ are key parameters that influence both approximation accuracy and memory usage. We suggest setting $k_{\min} = 1$ to fully leverage the low-rank structure of the second moment matrix. Similarly, setting $k_0 = k_{\max}$ helps effectively control the upper bound of memory

Table 7: Analysis on $p$, where $A$ is an $m \times n$ matrix sampled from a normal distribution ($m = 1024$, $n = 1024$, $k = 256$, $l = 5$).

| $p$ | 1 | 2 | 3 | 4 | 5 | 6 | 7 | 8 | 9 | 10 |
|---|---|---|---|---|---|---|---|---|---|---|
| $\|A - QU^\top\|_F / \|A\|_F$ | 0.0804 | 0.0797 | 0.0805 | 0.0807 | 0.0791 | 0.0797 | 0.0795 | 0.0803 | 0.0794 | 0.0799 |

consumption. The exact values for $k_0$ and $k_{\max}$ should be determined based on the available memory resources in practice.

Thank you for your suggestion. Below, we provide both a theoretical analysis and experimental results regarding memory utilization. We will incorporate this information into the revised manuscript accordingly.

## C ANALYSIS OF MEMORY USAGE FOR COMPRESSED SECOND MOMENT

In low-rank approximation for the second moment matrix, memory utilization is dominated by the storage of the decomposed matrices $Q \in \mathbb{R}^{m \times k}$ and $U \in \mathbb{R}^{n \times k}$. For a second moment matrix $V \in \mathbb{R}_{\geq 0}^{m \times n}$, the memory usage under low-rank approximation is:

$$\text{Memory Usage} = k \cdot (m + n),$$

where $k$ is the rank of the approximation. In contrast, storing the full matrix $V$ requires $m \times n$ elements. The ratio of memory saved can be expressed as:

$$\text{Memory Savings} = 1 - \frac{k \cdot (m + n)}{m \cdot n}.$$

For larger models (with higher $m$ and $n$), the relative savings increase as long as $k$ remains small compared to $m$ and $n$. However, increasing $k$ leads to higher memory usage, and for sufficiently large $k$, the savings diminish. For example, consider a second moment matrix $V$ of $1024 \times 1024$ ($m = n = 1024$):

- With $k = 16$, the memory savings are $1 - 16 \cdot (1024 + 1024)/1024^2 = 96.88\%$.
- With $k = 256$, the memory savings are $1 - 256 \cdot (1024 + 1024)/1024^2 = 50.00\%$.

In addition, to achieve memory savings in this example, $k$ must be less than $512$.

### C.1 EXPERIMENTAL RESULTS

To validate the efficiency, we measured the memory utilization of optimizer states for various ranks $k$ across GPT-2 models of different sizes during active training. The memory usage includes both the low-rank approximation and other optimizer states (e.g., first moment). Below are the results (in MB):

Table 8: Memory utilization (in MB) of optimizer states across GPT-2 models for various ranks $k$.

| Model | $k = 1$ | $k = 8$ | $k = 16$ | $k = 32$ | $k = 64$ | $k = 128$ | $k = 256$ | **AdamW** |
|---|---|---|---|---|---|---|---|---|
| GPT-2 125M | 475.93 | 481.27 | 487.39 | 499.61 | 524.06 | 572.95 | 670.74 | 949.40 |
| GPT-2 345M | 1356.47 | 1368.40 | 1382.02 | 1409.28 | 1463.79 | 1572.81 | 1790.85 | 2707.09 |
| GPT-2 1.1B | 3550.78 | 3573.88 | 3600.29 | 3653.09 | 3758.69 | 3969.90 | 4392.31 | 7089.92 |

As shown in the results above, when $k$ is small (e.g., 1 or 8), memory utilization for optimizer states is reduced by nearly half compared to AdamW, demonstrating significant savings due to compressed second moments. This reduction is particularly beneficial for larger models, such as GPT-2 1.1B.

As $k$ increases, memory usage grows, approaching that of AdamW at higher ranks. For example, at $k = 256$, memory utilization for GPT-2 1.1B is 38% lower than AdamW. This underscores the importance of selecting an appropriate $k$ to balance memory savings and approximation accuracy.

These results highlight the efficiency of low-rank approximations in reducing memory utilization for optimizer states, particularly when $k$ is small.

