# OpenReview forum: "Adapprox: Memory Efficient Optimization via Adaptive Randomized Low-Rank Approximation"
_ICLR.cc/2025/Conference — Submitted to ICLR 2025_

### Official Review · Reviewer_A5Fg · 2024-10-30

**Soundness:** 3
**Presentation:** 4
**Contribution:** 3
**Rating:** 8
**Confidence:** 3

**Summary:**

The paper presents a new optimizer ADAPPROX that can lead to a reduction of memory requirement needed to train a neural network.  The optimizer is tested on BERT and GPT2 pretraining tasks and achieved comparable performance with AdamW.

**Strengths:**

The paper presents an interesting approach to overcome the memory limitation of Adam and AdamW optimizer, especially related to training large language models.

The paper presents sufficient details related to the algorithm itself, making the paper easy to follow and comprehensive.

The paper's approach, to me, is novel and the impacts are valuable.

**Weaknesses:**

The experiment setting could be more comprehensive to illustrate the benefits, for example,  It would be great to have SGD baseline in the experiment section, to show how impactful the memory reduction is, as well as the training speed comparison.

**Questions:**

How will the proposed optimizer work in multi-node large scale training? Can author comment on whether there will be additional limitations or benefits of the approach in multi-node training scenario?

---

> ### Author Response · Authors · 2024-11-19
>
> **Response to Weaknesses**
>
> Thank you for your thoughtful suggestion. We appreciate your feedback and have carefully taken it into account.
>
> To address your concern, we conducted a series of new comparison experiments, including the baselines SGD and SGD with momentum (SGDM), as well as similar methods not previously included, such as GaLoreAdamW [1]. The training process was conducted from scratch on the GPT-2 125M model using the OpenWebText dataset. All experiments were performed on a single NVIDIA RTX 4090 GPU. The parameter settings for the compared methods are summarized in the table below.
> |**Method**|**Specific Parameters**|
> |-|-|
> |**Common**| Batch size = 8, Learning rate = 1e-4, Weight decay = 0.01, Steps=10K|
> |**SGD**|N/A|
> |**SGDM**|$\beta_1 = 0.9$|
> |**AdamW**|$\beta_1 = 0.9, \beta_2 = 0.999$|
> |**Adafactor**|$\beta_1 = 0.9, \beta_2 = 0.999$|
> |**CAME**|$\beta_1 = 0.9, \beta_2 = 0.999, \beta_3 = 0.9999$|
> |**GaLoreAdamW**|Rank = 128, Update proj gap = 200, Scale = 0.25, Proj type = std|
> |**Adapprox**|$k_0=k_{max}=k_{min}=1, l=1, p=5$|
>
> (*Those parameters not mentioned are set as default.*)
>
> We present the training loss recorded at every 1,000 steps along with the overall training time for each method in the table below:
> |**Method**|Step 1K|Step 2K|Step 3K|Step 4K|Step 5K|Step 6K|Step 7K|Step 8K|Step 9K|Step 10K|Time (min)|
> |-|-|-|-|-|-|-|-|-|-|-|-|
> |**SGD**|8.49|8.09|8.31|8.04|7.96|7.82|8.02|7.67|7.60|7.12|**22.49**|
> |**SGDM**|7.63|7.23|7.37|7.12|7.05|6.92|7.10|6.78|6.73|6.28|**22.71**|
> |**AdamW**|5.66|5.16|5.05|4.81|4.69|4.50|4.64|4.34|4.23|3.99|**23.81**|
> |**Adafactor**|5.64|5.15|5.04|4.80|4.69|4.50|4.64|4.35|4.24|4.00|**26.76**|
> |**CAME**|5.42|4.91|4.81|4.57|4.48|4.30|4.45|4.18|4.07|3.86|**29.14**|
> |**GaLoreAdamW (Full rank, Actually AdamW)**|5.67|5.18|5.07|4.82|4.70|4.51|4.65|4.36|4.24|4.01|**23.20**|
> |**GaLoreAdamW (rank=128)**|5.44|5.61|5.60|5.14|5.04|5.08|4.82|4.85|5.02|5.17|**24.97**|
> |**Adapprox (Ours)**|5.63|5.13|4.99|4.74|4.62|4.41|4.52|4.20|4.06|3.81|**28.84**|
>
> (*In the initial submission of this rebuttal, GaLoreAdamW's rank decomposition mechanism was not enabled, and the results corresponded to AdamW. We have now corrected this oversight and included the accurate data for rank = 128.*)
>
> The results above align with the conclusions presented in our manuscript. While SGD and SGDM require the least memory and computational cost, they converge significantly slower. This reinforces our belief that incorporating the second moment is essential for effectively training LLMs. CAME may reduce the training loss quickly in the initial stages but slows down as training progresses. Adafactor demonstrates convergence behavior similar to that of AdamW. As GaLoreAdamW compresses the gradient, which is crucial for guiding the optimization direction, its performance drops a lot and becomes unstable. Our Adapprox method demonstrates stable performance and achieves the lowest final training loss, even when the rank for approximation is aggressively fixed at a constant value of 1. This also indicates that, like Adafactor and CAME, we compress the second moment matrix using two vectors, effectively eliminating nearly all memory usage associated with the second moment.
>
> We will include these additional experimental results as figures and tables in our revised version to enhance the clarity and comprehensiveness of our experiments. Thank you again for your suggestion!
>
> [1] Zhao et al., 2024. GaLore: Memory-Efficient LLM Training by Gradient Low-Rank Projection.

---

> ### Author Response · Authors · 2024-11-19
>
> **Response to Question**
>
> Thank you for your question! Our method leverages low-rank matrix approximation to compress the second moment, reducing memory usage associated with storing  the second moment matrix. This compression not only benefits single-node training but also has notable implications in multi-node training:
>
> 1. In multi-node settings, where memory constraints are often exacerbated due to the large model sizes and batch splits, our approach retains its memory-saving advantage. By storing only low-rank approximations of second moments, we reduce the memory footprint across all nodes, enabling more efficient utilization of distributed resources.
> 2. Multi-node training typically involves communication of optimizer states (e.g., gradients, moments) between nodes. With our low-rank approximation, the second moment matrix representation is inherently compact, leading to reduced communication overhead compared to full-matrix methods. This could accelerate the synchronization process, especially in bandwidth-limited environments.
> 3. In particular, for tensor parallelism (e.g., splitting a large weight matrix across multiple nodes and thus the corresponding optimizer states), our "low-rank approximation" becomes even more accessible and applicable. Since the rank of a submatrix cannot exceed the rank of the original matrix (see detailed explanation below), applying low-rank approximation via Adapprox at each node for the corresponding submatrix has the potential to achieve higher accuracy.
>
> However, there are potential limitations to consider. Adapprox requires additional matrix operations during the S-RSI process. While these operations are efficient and scalable on modern hardware, in a distributed system, their execution time may slightly impact synchronization compared to simpler operations. Furthermore, if the synchronization interval is short (e.g., after every training step or a small number of steps), even minor latencies can accumulate over time, potentially increasing the overall training time.
>
> That said, there are viable mitigation strategies. For instance, reducing the frequency of low-rank matrix synchronization can help minimize latency. Additionally, performing low-rank projections in parallel with other computations (e.g., forward or backward passes) can effectively hide the associated latency.
>
> Overall, we believe our optimizer offers substantial benefits in multi-node training scenarios, particularly in terms of memory and communication efficiency. We will explore this aspect further in future work and include more detailed analysis in subsequent studies.
>
> (*Explanation for "the rank of a submatrix cannot exceed the rank of the original matrix"*)
>
> Without loss of generality, consider a large weight matrix $W$ used in a model, partitioned column-wise for tensor parallelism:
> $$
> W = [W_1 W_2 \cdots W_p].
> $$
>
> For each submatrix $W_i$, there is
> $$
> \mathrm{rank}(W_i)\leq \mathrm{rank}(W).
> $$
>
> This property follows directly from the definition of matrix rank: any set of linearly independent columns in $W_i$ must also be linearly independent in $W$. Therefore, the rank of $W_i$, being a subset of the columns of $W$, cannot exceed $\mathrm{rank}(W)$.

---

### Official Review · Reviewer_RuEU · 2024-11-04

**Soundness:** 2
**Presentation:** 2
**Contribution:** 2
**Rating:** 6
**Confidence:** 3

**Summary:**

This paper introduces Adapprox, a novel approach that leverages randomized low-rank matrix approximation to achieve a more effective and precise approximation of Adam’s second moment. The proposed method is more memory-efficient than Adam/AdamW and achieves greater accuracy than Adafactor/CAME due to its refined second-moment estimation. Additionally, the authors integrate a power iteration technique and an adaptive rank selection mechanism to further enhance the optimization process. Empirical studies on Transformer optimization are conducted, showcasing the effectiveness of the proposed method.

**Strengths:**

1. The integration of randomized low-rank matrix factorization into Adam optimization is an interesting idea, though its novelty remains uncertain.
2. Adapprox offers a flexible trade-off between memory usage and performance, which I consider to be an important contribution.
3. The paper is generally well-written, with a clear and easy-to-follow presentation.

**Weaknesses:**

1. Several design choices in this paper appear empirical and lack theoretical justification, such as the selection of $k_0$, $k_{min}$, $k_{max}$, $l$, and $p$. Additionally, these choices have neither been validated on larger-scale cases, such as optimizations for models with over 1 billion parameters, nor supported by any theory.
2. Certain techniques that are orthogonal to low-rank matrix factorization, such as CAME or quantization, are not explored in combination with the proposed method.

**Questions:**

See weaknesses.

---

> ### Author Response · Authors · 2024-11-22
> **Part 1**
>
> **Response to Weaknesses 1**
>
> Thank you for your thoughtful comment. The parameters $k_0$, $k_{min}$, $k_{max}$, $l$, and $p$ are indeed selected based on the trade-offs between approximation accuracy, memory usage, and computation efficiency.
>
> In terms of approximation accuracy,  Equation (10) in our manuscript illustrates how these parameters influence the low-rank approximation:
> $$
> \mathbb{E} \| A - QU^\top\| \leq \left[ \left( 1 + \sqrt{\frac{k}{p - 1}} \right)^{2l+1} \sigma_{k+1}^{2l+1} \right.
>      + \frac{e\sqrt{k + p}}{p} \left. \sqrt{\sum_{j > k} \sigma_j^{2(2l+1)}} \right]^{1/(2l+1)}.
> $$
>
> As shown, larger values of $k, l$ and $p$ generally result in smaller approximation errors. However, increasing these parameters also results in higher computational cost or memory consumption.
>
> First, $l$ represents the number of power iterations in our S-RSI method. While increasing $l$ improves approximation accuracy, it also inherently increases computation cost, as power iterations are sequential. Based on our experiments, $l=5$ offers a good balance between accuracy and efficiency. Below are the results of an ablation study on $l$, where $A$ is a $m \times n$ matrix sampled from a normal distribution:
>
> |$m=1024, n=1024, k=256, p=5$|l=1 |l=2|l=3|l=4|l=5|l=6|l=7|l=8|l=9|l=10|
> |-|-|-|-|-|-|-|-|-|-|-|
> |$\|\|A-QU^T\|\|_F/\|\|A\|\|_F$|0.1473|0.0968|0.0850|0.0828|0.0802|0.0786|0.0783|0.0779|0.0779|0.0778|
>
> As shown, $l=5$ achieves sufficient accuracy, with further increases yielding only marginal improvements. Therefore, we recommended $l=5$ in our manuscript.
>
> Similarly, we recommended $p=5$ in our manuscript:
>
> |$m=1024, n=1024, k=256, l=5$|p=1 |p=2|p=3|p=4|p=5|p=6|p=7|p=8|p=9|p=10|
> |-|-|-|-|-|-|-|-|-|-|-|
> |$\|\|A-QU^T\|\|_F/\|\|A\|\|_F$|0.0804|0.0797|0.0805|0.0807|0.0791|0.0797|0.0795|0.0803|0.0794|0.0799|
>
> Besides, $k_0$, $k_{max}$, and $k_{min}$ are critical parameters that govern both approximation accuracy and memory usage.  We suggest setting $k_{min} = 1$ to ensure that our method could fully exploit the low-rank structure of the second moment matrix. Meanwhile, we recommend setting $k_0 = k_{max}$ to effectively control the upper bound of memory consumption. The specific values for $k_0$ and $k_{max}$ depend on the available memory resources in practice.
>
> Regarding the need to validate our design choices on larger-scale cases. To address this, we conducted additional experiments on a 1.1B parameter GPT-2 model using 4 NVIDIA RTX 4090 GPUs. The model configuration is as follows:
>
> - n_positions = 1024
> - n_ctx = 1024
> - n_embd = 1408
> - n_layer = 36
> - n_head  = 22
>
> The parameter setting is as follows:
> |**Method**|**Specific Parameters**|
> |-|-|
> |**Common**| Batch size = 8, Learning rate = 1e-4, Weight decay = 0.01, Steps=10K|
> |**AdamW**|$\beta_1 = 0.9, \beta_2 = 0.999$|
> |**Adafactor**|$\beta_1 = 0.9, \beta_2 = 0.999$|
> |**CAME**|$\beta_1 = 0.9, \beta_2 = 0.999, \beta_3 = 0.9999$|
> |**Adapprox**|$k_0=k_{max}=256, k_{min}=1, l=5, p=5$|
>
> (*Those parameters not mentioned are set as default.*)
>
> The results are as follows:
>
> |**Method**|Step 1K|Step 2K|Step 3K|Step 4K|Step 5K|Step 6K|Step 7K|Step 8K|Step 9K|Step 10K|
> |-|-|-|-|-|-|-|-|-|-|-|
> |**AdamW**|5.269|4.585|4.485|4.279|4.353|4.520|4.219|4.438|3.825|3.605|
> |**Adafactor**|5.251|4.562|4.457|4.258|4.324|4.499|4.202|4.411|3.809|3.597|
> |**CAME**|5.306|4.687|4.675|4.481|4.563|4.751|4.445|4.713|4.093|3.896|
> |**Adapprox (Ours)**|5.223|4.536|4.456|4.249|4.319|4.502|4.202|4.411|3.810|3.592|
>
> As shown, our parameter choices remain effective for Adapprox even on larger models. We will include these additional results in the revised manuscript to address the concern and further demonstrate the robustness of our approach.
>
> In practice, optimal hyperparameter settings may be different on the specific use case. As indicated in Equation (10), increasing $p$, $l$, and $k$ reduces approximation error but also will increase memory and computational cost. Therefore, we recommend a trade-off based on the available hardwares and the requirements of the training tasks.

---

> > ### Comment · Reviewer_RuEU · 2024-11-25
> >
> > Thank you for your response. I appreciate the effort the authors made to conduct additional experiments. I am convinced by the justification for choosing p = 5 and l = 5. However, in the larger case (1.1B GPT-2), I observed that the loss curve of Adapprox is highly similar to that of Adafactor. Could you please provide further explanation for this observation?
> >
> > According to my understanding, Adapprox should consistently outperform Adafactor. If such performance gains cannot be achieved with minimal hyperparameter tuning, it may suggest that Adapprox is less practical in real-world applications.

---

> > > ### Author Response · Authors · 2024-11-26
> > >
> > > Thank you for your thoughtful feedback and for acknowledging the additional experiments we conducted. We have included an analysis of the selection of $p$ and $l$ in Appendix. Besides, we sincerely appreciate the opportunity to address your observations.
> > >
> > > The similarity between the loss curves of Adapprox and Adafactor in our previous GPT-2 1.1B setting can be attributed to the following factors:
> > >
> > > - **Task and Model Characteristics**:
> > >   Due to our current hardware limitations and the constrained response time, we conducted experiments with GPT-2 1.1B using a batch size of 8, which is smaller than typically used in practical applications. In this setup, the 1.1B GPT-2 model represents a large and overparameterized system.  Such models are often robust to optimization differences because their size allows them to fit training data effectively. Such models often exhibit robustness to optimization differences because their size enables them to fit training data effectively. This makes the optimization landscape relatively forgiving, allowing even low-rank approximations like the rank-1 approach to perform well.
> > >   To validate this point, we retained the original parameters for the optimizers and increased the batch size to 12 using the same training data (and thus about 6K iterations). The results are as follows:
> > >   |Method|Step 600|Step 1200|Step 1800|Step 2400|Step 3000|Step 3600|Step 4200|Step 4800|Step 5400|Step 6000|
> > >   |-|-|-|-|-|-|-|-|-|-|-|
> > >   |Adafactor|5.51|5.33|5.11|5.23|4.76|4.42|4.12|4.11|4.00|3.72|
> > >   |Adapprox|5.40|5.24|5.10|5.23|4.74|4.40|4.09|4.08|3.99|3.70|
> > >
> > >   These results show that in a slightly more challenging setting, Adapprox's performance advantage becomes more apparent. However, Adafactor can exhibit faster training loss reduction at certain stages due to the highly non-convex optimization landscape of large language models. In such cases, noise or fluctuations from less precise approximations (e.g., rank-1) may occasionally help escape local minima. Despite this, over the entire training process, a more precise approximation of the second moment typically leads to greater stability and better convergence. This aligns with why AdamW, which uses full second moments, is often preferred in applications.
> > > - **Hyperparameter Tuning**
> > >   In our previous experiments, we used a consistent set of hyperparameters ($k_0=k_{max}=256, k_{min}=1, \xi=1 \times 10^{-3}$) for Adapprox as outlined in our paper, to ensure fairness and simplicity. However, these settings may not fully exploit the potential of Adapprox in the current GPT-2 1.1B set up. Prior results indicate that the peak rank $k_{max}$ may not need to be as large as 256. Besides, $k$ may be adjusted too aggressively. To further investigate this, we adjusted $k_{max}$ to 16 and the error threshold $\xi$ to $5\times 10^{-4}$. The results are as follows:
> > >   |Method|Step 600|Step 1200|Step 1800|Step 2400|Step 3000|Step 3600|Step 4200|Step 4800|Step 5400|Step 6000|
> > >   |-|-|-|-|-|-|-|-|-|-|-|
> > >   |Adafactor|5.51|5.33|5.11|5.23|4.76|4.42|4.12|4.11|4.00|3.72|
> > >   |Adapprox ($k_0=k_{max}=256, k_{min}=1, \xi=1\times 10^{-3}$)|5.40|5.24|5.10|5.23|4.74|4.40|4.09|4.08|3.99|3.70|
> > >   |Adapprox ($k_0=k_{max}=16, k_{min}=1, \xi=5\times 10^{-4}$)|5.38|5.22|5.07|5.19|4.72|4.39|4.09|4.08|3.97|3.69|
> > >
> > >   These results demonstrate that the updated parameter settings not only improve convergence but also reduce memory usage and computational costs dur to smaller rank value.
> > >
> > >   Therefore, in practice, the parameters of Adapprox can be selected or tuned based on the principle of balancing approximation accuracy and memory usage. This flexibility makes Adapprox adaptable to diverse applications and hardware constraints.
> > >
> > >   In summary, Adapprox is designed to provide a dynamic low-rank approximation for the second moment, with Adafactor serving as the expected performance baseline for the case where $k=1$.

---

> > > > ### Comment · Reviewer_RuEU · 2024-11-27
> > > >
> > > > Thank you for the further clarification. It's good to see that some performance gains can be achieved in more challenging cases with appropriate hyperparameter tuning. I encourage the authors to evaluate Adapprox in additional practical scenarios, particularly with larger models, larger batch sizes, or increased numbers of cumulation steps.
> > > >
> > > > I have decided to increase my rating to 6 and my confidence to 3, as I support the acceptance of this paper. The reason I haven't increased my rating and confidence further is the lack of evaluations in more practical scenarios.

---

> > > > > ### Author Response · Authors · 2024-11-29
> > > > >
> > > > > Thank you for your thoughtful comments and for increasing your rating and confidence in our work. We truly appreciate your encouragement and the suggestion to evaluate Adapprox in additional practical scenarios. We will consider expanding the evaluation in future work to include these scenarios. Thanks again for your valuable feedback and support!

---

> ### Author Response · Authors · 2024-11-22
> **Part 2**
>
> **Response to Weakness 2**
>
> Thank you for your insightful comment. We agree that techniques orthogonal to low-rank matrix factorization, such as CAME or quantization, have significant potential to complement our proposed method. While our primary focus was to develop and analyze the core low-rank approximation approach, we recognize the value of integrating these techniques to further enhance efficiency and scalability.
>
> In particular, CAME inspire us to develop additional mechanisms to address approximation errors in the low-rank approximation of the second moment matrix. Combining our method with quantization could reduce memory usage and computational overhead further. Exploring such synergies is a promising direction for future work, and we will consider this in subsequent research. We appreciate your suggestion and will include a discussion of this point in the manuscript to highlight these opportunities.

---

### Official Review · Reviewer_ytkG · 2024-11-07

**Soundness:** 3
**Presentation:** 3
**Contribution:** 3
**Rating:** 8
**Confidence:** 4

**Summary:**

This paper introduces Adapprox, a memory-efficient optimizer designed to address memory consumption challenges in large-scale model training. It uses an adaptive randomized low-rank approximation for the second moment.

Problem & Motivation: Large models like GPT-3 and BERT demand substantial memory due to optimizers such as Adam, which store both first and second moments. Existing memory-efficient methods (e.g., Adafactor and CAME) compromise accuracy by relying on constant rank-1 approximations. Adapprox is proposed to retain performance while minimizing memory.

Methodology: The method leverages low-rank characteristics in second-moment matrices, reducing memory usage without excessive computation costs. Instead of fixed-rank approximations, Adapprox uses an adaptive rank that adjusts dynamically based on iteration needs. The rank is adjusted according to an error threshold, and an exponential averaging mechanism stabilizes updates. Adapprox balances memory savings with approximation accuracy by computing approximations using the Streamlined Randomized Subspace Iteration (S-RSI) method.

Experiments & Results: Adapprox was tested with models like GPT-2 and BERT, showing significant memory reductions and superior performance (in terms of convergence and validation loss) over AdamW, Adafactor, and CAME. Fine-tuned models trained with Adapprox achieved high accuracy across NLP tasks, outperforming other memory-efficient optimizers. Adapprox incurred a modest increase in latency (~6%) but maintained significant memory savings.

Conclusions:
Adapprox is presented as a balanced solution that provides substantial memory efficiency and high performance in model training. The paper suggests future work on reducing the frequency of rank adjustments to optimize computational efficiency further. This optimizer is positioned as a promising approach to training large models with less memory and minimal performance trade-offs compared to state-of-the-art optimizers.

**Strengths:**

Originality:
The paper presents Adapprox, an optimizer that utilizes adaptive low-rank approximation for the second moment in large-scale model training. This method flexibly modifies the rank throughout the training process, achieving a balance between memory efficiency and accuracy. This flexibility sets Adapprox apart from static methods such as Adafactor and CAME. The combination of Streamlined Randomized Subspace Iteration (S-RSI) and Adaptive Rank Selection (AS-RSI) algorithms improves both computational efficiency and precision in low-rank approximations, showcasing the approach's novelty.

Quality:
Thorough experiments on models like GPT-2 and BERT validate this approach, showing notable memory savings and better convergence rates against optimizers such as AdamW, Adafactor, and CAME. The paper includes in-depth assessments of memory utilization, convergence patterns, and performance on downstream tasks, providing strong proof of Adapprox’s efficacy. The technical descriptions of S-RSI and AS-RSI are clearly articulated, accompanied by understandable pseudocode, which enhances reproducibility and comprehension.

Clarity:
The paper features a clear structure, progressing logically from problem motivation through methodology and experiments to conclusions. Technical concepts are articulated effectively and supported by equations and pseudocode that facilitate understanding. The experimental setup is thoroughly detailed, while the results are illustrated with informative figures and tables that help clarify the findings.

Significance:
Adapprox tackles a significant issue in training large-scale models by lowering memory usage while maintaining performance, greatly enhancing its relevance in the field. The optimizer's adaptive characteristics enable it to respond to different training conditions, which could result in more efficient training methods. This strategy has real-world implications for implementing large models in resource-limited settings, expanding the reach of advanced AI technologies.

**Weaknesses:**

Limited Comparison with Adaptive Rank Techniques:
Although Adapprox presents a method for adaptive low-rank approximation, the paper lacks a comprehensive comparison with other adaptive rank optimization techniques. A detailed examination against recent studies on adaptive low-rank or memory-efficient methods that utilize varying ranks would enhance the originality and rigor of this work.
An in-depth examination of the latest developments in adaptive approximation methods and low-rank optimization would offer readers clearer insights into how Adapprox compares to existing techniques beyond just Adafactor and CAME.

Complexity in Explanation of Adaptive Mechanism:
The explanation of the Adaptive Rank Selection (AS-RSI) mechanism would benefit from condensing and providing extra context, particularly for those not familiar with randomized SVD or subspace iteration. While it's organized well, the technical specifics about rank adjustment thresholds and error ratios could be clarified with a more intuitive or overarching summary before presenting the equations and pseudocode. Making this section more approachable could involve adding a flowchart or offering a simplified, step-by-step outline of AS-RSI’s rank adaptation process.

Analysis of Latency and Efficiency Trade-offs:
Adapprox shows remarkable memory efficiency, accompanied by only a slight rise in latency; however, this latency-memory trade-off warrants a more detailed analysis. Investigating how latency increases with larger ranks or deeper architectures would shed light on the optimizer’s capabilities in practical scenarios. Additionally, given the results indicate a slight increase in latency with Adapprox, breaking down the computational costs associated with S-RSI and AS-RSI components could help identify which factors have the greatest impact on the additional time cost.

Impact of Rank Adaptation on Convergence:
Further investigation into how rank adaptation affects convergence rates, especially in relation to different error thresholds and AS-RSI parameters, is warranted. Although the experiments show advantages in convergence, examining the impact of rank selection on convergence stability would provide additional insights. Conducting an ablation study that alters error thresholds and monitors convergence speed alongside final model performance would also be beneficial. This type of analysis could provide clear guidance for practitioners seeking to optimize Adapprox’s configurations according to their memory and accuracy requirements.

Scope of Downstream Tasks:
The downstream tasks tested are well-chosen for NLP, but expanding to other domains (e.g., vision tasks or different LLM architectures) could strengthen claims about Adapprox’s generalizability. Given the optimizer’s potential for a wide range of models, testing on a broader selection of model types could emphasize its applicability beyond the NLP-focused experiments.

**Questions:**

Comparison with Adaptive Rank Methods:
Can you elaborate on how Adapprox contrasts with modern adaptive-rank or low-rank approximation techniques regarding memory-efficient optimization? For example, approaches like GaLORE and CAME utilize low-rank strategies but depend on a fixed rank. Are there other adaptive-rank methods besides these that you’ve considered or think are pertinent to your methodology? Additionally, a more in-depth comparison with GaLORE, which emphasizes memory-efficient optimization via low-rank gradient projections, would clarify Adapprox's distinctive contributions.

Clarification on Adaptive Rank Selection (AS-RSI):
The adaptive rank selection mechanism within AS-RSI is crucial for Adapprox’s adaptability. Can you elaborate on how the error threshold and rank modifications are established? For instance, is there an empirical method for selecting threshold values, or are they adjusted specifically for each model?

Exploration of Latency and Memory Trade-off:
Considering the minor increase in latency with Adapprox, could you share more detailed insights on how various algorithm components, such as S-RSI and AS-RSI, add to this overhead? Analyzing the added computational time could help us pinpoint potential optimization areas. Would you be willing to run experiments with different model sizes or ranks to investigate how latency varies and better understand the algorithm’s effectiveness for diverse large models?

Convergence Analysis and Ablation Studies:
Can you elaborate on how rank adaptation influences convergence? Specifically, how responsive are the outcomes to changes in the error threshold applied in AS-RSI? Conducting an ablation study that varies error thresholds and demonstrates their effects on memory efficiency and convergence stability would bolster your assertions regarding the effectiveness of the adaptive rank. Would you like to include this to substantiate AS-RSI’s adaptive mechanism further?

Availability of Code for Reproducibility:
Will you provide code to support reproducibility? If yes, can you suggest specific guidelines or configuration settings for those looking to replicate your results, especially concerning rank selection and error thresholds?

---

> ### Author Response · Authors · 2024-11-23
> **Part 1**
>
> Thank you for your thoughtful and detailed feedback, as well as your positive evaluation of our work. Below, we address your insightful suggestions and provide further clarification on the points you raised.
>
> **Response to Weakness 1 and Question 1**
>
> Thank you for your comment.
>
> ***Comparison with Adaptive Rank Techniques***
>
> The compared methods, Adafactor [1] and CAME [2], rely on fixed rank-1 low-rank approximation techniques and do not include an adaptive rank mechanism. To the best of our knowledge, our work is the first to explicitly introduce an adaptive rank mechanism in memory-efficient optimization algorithms via low-rank matrix approximation.
>
> The motivation behind this innovation is that our S-RSI enables flexible rank approximation, allowing the algorithm to dynamically select the rank for approximating the second-moment matrix as needed. This adaptability ensures a balance between approximation accuracy, computational cost, and memory utilization.
>
> ***Comparison with GaLore***
>
> GaLore [3] emphasizes memory efficiency through low-rank gradient projections, which target gradient compression (and consequently both the first and second moments), rather than focusing solely on the second moment as Adapprox does. Considering its mechanism, GaLore offers both potential benefits and limitations compared to our methods:
>
> - *Benefits:* GaLore theoretically enables higher memory savings. While Adapprox (along with Adafactor and CAME) focuses solely on compressing the second moment, which provides a memory saving upper bound equivalent to the storage of the second moment, GaLore projects gradients into a compact space. This allows for memory savings with respect to both the first and second moments, potentially achieving greater overall memory efficiency. In addition, by projecting gradients into a compact space, GaLore also achieves greater computational efficiency in the corresponding operations.
> - *Limitations:* While GaLore compresses gradients to achieve memory efficiency, gradients are critical as they provide the direction for updates during optimization. Any inaccuracies in the gradient direction could lead to instability in the optimization trajectory, potentially affecting both convergence speed and overall performance. Therefore, this makes it more challenging to achieve a suitable trade-off between memory utilization and training performance.
>
> To compare GaLore with our method in depth, we conducted a series of experiments over GPT-2 125M on NVIDIA RTX 4090 GPU. The configurations for the experiments are as follows:
>
> |**Method**|**Specific Parameters**|
> |-|-|
> |**Common**|Batch size=8, Learning rate=1e-4, Weight decay=0.01, Steps=10K|
> |**AdamW**|$\beta_1 = 0.9, \beta_2 = 0.999$|
> |**GaLoreAdamW**|Rank = [1, 128, 256, 512], Update proj gap = 200, Scale = 0.25, Proj type = std|
> |**Adapprox**|$k_0=k_{max}=k_{min}=1, l=1, p=5$|
>
>
> The recorded training losses at every 1,000 steps are presented below:
>
> |**Method**|Step 1K|Step 2K|Step 3K|Step 4K|Step 5K|Step 6K|Step 7K|Step 8K|Step 9K|Step 10K|Time (min)|
> |-|-|-|-|-|-|-|-|-|-|-|-|
> |**AdamW**|5.66|5.16|5.05|4.81|4.69|4.50|4.64|4.34|4.23|3.99|**23.81**|
> |**GaLoreAdamW (rank=1)**|5.84|5.85|5.77|5.34|5.24|5.29|5.04|5.07|5.24|5.39|**24.57**|
> |**GaLoreAdamW (rank=128)**|5.44|5.61|5.60|5.14|5.04|5.08|4.82|4.85|5.02|5.17|**24.97**|
> |**GaLoreAdamW (rank=256)**|5.41|5.58|5.52|5.10|4.99|5.02|4.75|4.78|4.94|5.09|**25.35**|
> |**GaLoreAdamW (rank=512)**|5.38|5.55|5.49|5.06|4.93|4.94|4.67|4.67|4.84|4.99|**25.47**|
> |**Adapprox (rank=1)**|5.63|5.13|4.99|4.74|4.62|4.41|4.52|4.20|4.06|3.81|**28.84**|
>
> The memory utilization are presented below:
>
> |Method|k=1|k=128|k=256|
> |-|-|-|-|
> |Adapprox|475.93|481.27|487.39|499.61|
> |GaLoreAdamW|302.24|409.40|517.40|
>
> The results above align well with our prior analysis. GaLoreAdamW demonstrates greater computational efficiency and could achieve higher memory savings for the same rank setting. However, Adapprox achieves better and more stable training performance, showcasing its robustness in optimization tasks. (Notably, when memory utilization is comparable, such as $k=1$ for Adapprox and $k=256$ for GaLoreAdamW, our Adapprox outperforms GaLoreAdamW.)
>
>
> [1] Shazeer, Noam, and Mitchell Stern. "Adafactor: Adaptive learning rates with sublinear memory cost." International Conference on Machine Learning. PMLR, 2018.
>
> [2] Luo, Yang, et al. "CAME: Confidence-guided Adaptive Memory Efficient Optimization." Proceedings of the 61st Annual Meeting of the Association for Computational Linguistics (Volume 1: Long Papers). 2023.
>
> [3] Zhao, Jiawei, et al. "GaLore: Memory-Efficient LLM Training by Gradient Low-Rank Projection." Forty-first International Conference on Machine Learning.

---

> ### Author Response · Authors · 2024-11-23
> **Part 2**
>
> **Response to Weakness 2 and Question 2**
>
> Thanks for your question. We define the approximation error as $||A-QU^\top||_F/||A||_F$, which measures the normalized error ratio. Since this metric captures the relative error, it is minimally influenced by the size or magnitude of $A$, making it adaptable to models of varying sizes.
>
> Regarding the specific value of the threshold $\xi$ for $||A-QU^\top||_F/||A||_F$, which controls the adaptive rank direction (whether to increase or decrease the rank), we currently set it empirically to $1\times 10^{-3}$. This value was selected after testing several options within the range $[1 \times 10^{-5}, 1 \times 10^{-2}]$, as it provides a good balance between accuracy and rank adjustment. A threshold that is too small results in slower rank reduction and increased computational cost, while a threshold that is too large reduces the rank too aggressively, potentially impacting accuracy.
>
> For rank modifications, we were inspired by the gradient descent framework and designed a rank adjustment mechanism in the form $k \pm \Delta k$. This approach allows $k$ to adapt incrementally within a local neighborhood, enabling the algorithm to maintain approximation performance comparable to the previous step while effectively exploring and adjusting the rank. Currently, we empiriclly set $\Delta k$ as $\lceil 0.02k_{max} \rceil$.
>
> **Response to Weakness 3 and Question 3**
>
> Thank you for your question.
>
> First, the ratio of additional latency introduced by Adapprox (due to S-RSI or AS-RSI) over the overall latency during training depends on factors such as the model size, batch size (which affects forward pass time), and the hardware used. Below, we provide detailed record for GPT-2 117M trained on 8 NVIDIA Tesla V100 GPUs with a global batch size of 128 and a sequence length of 1024. The timings were recorded using TensorBoard’s PyTorch Profiler, with the start of the step set to 0 seconds:
>
> |Phase|Time Duration|Time Cost|
> |-|-|-|
> |Forward Pass|0.000-0.402s|0.402s|
> |Loss Computation and others|0.402-0.412s|0.010s|
> |Backward Pass|0.412-1.742s|1.330s|
> |Device Synchronize and others|1.742-1.806s|0.064s|
> |Adapprox Optimizer Computation|1.806-2.046s|0.240s|
> |-|-|-|
> |Total|0.000-2.046s|2.046s|
>
> As results shown above, the backward pass constitutes the majority of the training time. From this perspective, although Adapprox introduces additional computations, the relative increase in time remains acceptable.
>
> We also conducted experiments with different model sizes and ranks to investigate how latency varies. Due to limitations in our current hardware and the constrained rebuttal period, we used 4 NVIDIA RTX 4090 GPUs with a batch size of 8. For this study, we selected GPT-2 models with 125M, 345M, and 1.1B parameters. The rank was fixed at 8, 16, 32, 64, 128, and 256, respectively. We recorded the training speed (iterations per second) for each rank and the total training time over 1,000 steps. The parameters for S-RSI were set to $p=5$ and $l=5$.
>
> |**Model**|**Rank ($k$)**|**Training Speed (iter/sec)**| **Total Training Time (min:sec)**|
> |-|-|-|-|
> |**GPT-2 125M**|$k=8$|4.84|3:26|
> ||$k=16$|4.74|3:31|
> ||$k=32$|4.77|3:29|
> ||$k=64$|4.69|3:33|
> ||$k=128$|3.89|4:17|
> ||$k=256$|2.53|6:35|
> |**GPT-2 345M**|$k=8$|2.46|6:45|
> ||$k=16$|2.42|6:53|
> ||$k=32$|2.40|6:56|
> ||$k=64$|2.31|7:12|
> ||$k=128$|1.72|9:41|
> ||$k=256$|1.08|15.26|
> |**GPT-2 1.1B**|$k=8$|1.34|12:24|
> ||$k=16$|1.32|12:36|
> ||$k=32$|1.31|12:41|
> ||$k=64$|1.16|14:19|
> ||$k=128$|1.13|18:46|
> ||$k=256$|0.58|28:47|
>
> Preliminary findings suggest that while higher ranks lead to increased computation time, the scaling remains sub-linear due to efficient parallelization and the overlap of computations with other training tasks. When using the same rank $k$ across models of different sizes,the number of layers must be considered, since layers are processed sequentially. For example, GPT-2 125M has 12 layers, while GPT-2 345M has 24 layers.
>
> Note that the results above represent a relative comparison within our specific setting. Furthermore, as the batch size and number of devices increase, the proportion of additional time introduced by Adapprox's S-RSI or AS-RSI relative to the total training time will be further reduced.

---

> ### Author Response · Authors · 2024-11-23
> **Part 3**
>
> **Response to Weakness 4 and Question 4**
>
> Thank you for your question.
>
> To elaborate on the influence of rank adaptation on convergence, it is essential to revisit the intuition and motivation behind our approach. Our goal is to provide a more flexible method for approximating the second moment during training, balancing the trade-off between training performance, memory usage, and computational cost.
>
> The underlying intuition is that higher accuracy in the approximation (achieved by using a higher rank $k$) results in behavior more closely aligned with the counterpart optimizer, AdamW. Consequently, a higher rank $k$ typically leads to more stable convergence during training.
>
> It is important to emphasize the term "stable" here, as optimizing large language models involves navigating a highly non-convex landscape. In such scenarios, occasional noise or fluctuations (resulting from less precise approximations) can sometimes help escape local minima and achieve better training loss, highlighting the complex interplay between accuracy and optimization dynamics.
>
> As for our rank adaption mechanism, the error threshold $\xi$ in AS-RSI plays a key role in controlling rank adjustments. A smaller $\xi$ tends to maintain higher ranks for longer, ensuring greater accuracy but at the expense of higher computational overhead. Conversely, a larger $\xi$ leads to more aggressive rank reductions, potentially compromising convergence stability.
>
> To demonstrate the influence of rank adaption mechanism and the choice of $\xi$ on convergence performance, we conducted the following ablation studies on GPT-2 345M using 4 Nvidia RTX 4090 GPUs and with a batch size of 8 over 1,000 steps. We tested three scenarios:
> - (1) Fixing $k=128$ to represent a case where $\xi$ is too small;
> - (2) Fixing $k=1$ to represent a case where $\xi$ is too large;
> - (3) Using $\xi=1\times 10^{-3}, k_0=k_{max}=128, k_{min}=1$, which represents the adaptive rank mechanism.
>
> The training loss results are as follows:
>
> |Cases|Step 200|Step 400|Step 600|Step 800|Step 1000|
> |-|-|-|-|-|-|
> |Case 1 (k=128)|6.075|5.739|5.592|5.552|5.408|
> |Case 2 (k=1)|6.16|5.781|5.619|5.586|5.441|
> |Case 3 (adaptive)|6.075|5.739|5.595|5.565|5.413|
>
> The results above align with our analysis. The adaptive rank mechanism achieves performance between the two extremes, demonstrating its ability to balance accuracy, memory efficiency, and computational cost. The mechanism could effectively adjust the rank to meet the demands of each training stage, avoiding unnecessary computational overhead while maintaining stable convergence.
>
> **Response to Weakness 5**
>
> Thanks for your comment. Our current focus on NLP downstream tasks aligns with prior work in the field and reflects the natural applicability of Adapprox to transformer-based architectures. We agree that extending Adapprox to other domains, such as vision tasks or alternative model architectures, could further strengthen its generalizability. The adaptive rank mechanism and memory-efficient optimization strategies used in Adapprox are not inherently tied to NLP models and could be adapted for other domains. We will explore these broader applications in future work.
>
> **Response to Question 5**
>
> Thanks for your question. We  will release it publicly upon acceptance of the manuscript. The detailed experimental settings are provided in Section 4 of the manuscript. Specifically, our choice for rank setting is $k_0=k_{max}=256$. The rank adjustment is $\Delta k = \lceil 0.02 k_{max} \rceil$. The error threshold is $\xi = 1\times 10^{-3}$.
>
> For specific use cases, we recommend adjusting these parameters based on the available memory and training requirements. Selecting an appropriate rank is especially crucial to achieve a good trade-off between memory savings and computational efficiency while maintaining training performance.

---

### Official Review · Reviewer_YMzP · 2024-11-08

**Soundness:** 2
**Presentation:** 3
**Contribution:** 2
**Rating:** 5
**Confidence:** 4

**Summary:**

This work proposes a strategy for reducing the memory consumed by the second moment buffer in Adam. Inspired by Adafactor, which computes a rank-1 approximation to the second-moment buffer matrices in Adam, this work computes a rank-r approximation using a randomized SVD, thereby achieving a desirable tradeoff between memory utilization and model performance. The decomposition rank, "r", is updated dynamically during training by simply measuring the approximation error of the low-rank factorization. Numerical results provided for GPT2 and BERT models, primarily comparing to Adafactor and CAME (an uncertainty aware variant of Adafactor).

**Strengths:**

1. Clarity: The paper is well written and easy to understand.
2. Clarity: The method is conceptually simple and well motivated.
3. Quality: Numerical results demonstrate promising performance, achieving strong performance with limited computational overhead due to computing a higher-rank approximation of the second moment buffers compared to Adafactor and CAME.
4. Significance: The paper is likely to be of interest to a broad audience, and tackles an important problem reducing the memory footprint of language model optimizers.

**Weaknesses:**

* Computation times for higher rank decompositions are quite flat in the ablations (Figure 2b and Table 1); this is attributed to parallelization on the GPU, but, how does this time actually scale during training when the GPU supposedly has high SM occupancy?
* It seems a little odd that all the validation curves in Figure 4 (including those using different optimizers), line up perfectly, down to every minute jump or dip in the loss and perplexity.

**Questions:**

* Please investigate issues in the training runs used to produce Figure 4.
* Please report the ablations examining training time with higher rank decompositions and various model sizes; the idea is to understand how the rank affects the decomposition time during training (not just on an idle GPU).
* Consider adding memory utilization and training time in Table 3 for all methods; this will provide some insight into the relative memory utilization between Adafactor and the proposed method.
* Consider including experiments demonstrating the effect of the rank decomposition R on memory utilization for the various model sizes.

Provided further confidence in the numerical performance of the method, I am willing to increase my score.

---

> ### Author Response · Authors · 2024-11-22
> **Part 1**
>
> **Response to Question 1 and Weakness 2**
>
> Thank you for your observation regarding the validation curves in Figure 4. The similarity in the validation curves arises because all experiments were conducted under identical training conditions, including model initialization, data ordering, and random seeds. This ensures a controlled comparison and eliminates variability that could obscure the differences introduced by the optimizers. The jumps or dips in the curves reflect specific points in the training process where the model interacts with particularly challenging or easy batches, which remain consistent across all runs due to the shared conditions.
>
> Despite this, we also suspect that the optimization landscape is highly non-convex, characterized by large flat regions interspersed with dense clusters of local minima. This structure could contribute to the consistent behavior of the validation curves across different optimizers, as they may follow similar trajectories through the landscape under identical initialization and training conditions. We are interested in further exploring this suspicion in future work.
>
> **Response to Question 2 and Weakness 1**
>
> Thank you for your suggestion to examine the impact of rank on decomposition time during training. To address this, we conducted additional experiments measuring the training speed across different ranks and model sizes during active training on 4 NVIDIA RTX 4090 GPUs. For this study, we selected GPT-2 models with 125M, 345M, and 1.1B parameters. We fixed the rank at 8, 16, 32, 64, 128 and 256, respectively. We recorded the training speed (iteration per sections) for each rank and record the total training time over 1,000 steps. The parameters for S-RSI are $p=5$ and $l=5$.
>
>
> |**Model**|**Rank ($k$)**|**Training Speed (iter/sec)**| **Total Training Time (min:sec)**|
> |-|-|-|-|
> |**GPT-2 125M**|$k=8$|4.84|3:26|
> ||$k=16$|4.74|3:31|
> ||$k=32$|4.77|3:29|
> ||$k=64$|4.69|3:33|
> ||$k=128$|3.89|4:17|
> ||$k=256$|2.53|6:35|
> |**GPT-2 345M**|$k=8$|2.46|6:45|
> ||$k=16$|2.42|6:53|
> ||$k=32$|2.40|6:56|
> ||$k=64$|2.31|7:12|
> ||$k=128$|1.72|9:41|
> ||$k=256$|1.08|15.26|
> |**GPT-2 1.1B**|$k=8$|1.34|12:24|
> ||$k=16$|1.32|12:36|
> ||$k=32$|1.31|12:41|
> ||$k=64$|1.16|14:19|
> ||$k=128$|1.13|18:46|
> ||$k=256$|0.58|28:47|
>
> Preliminary findings suggest that while higher ranks lead to increased computation time, the scaling remains sub-linear due to efficient parallelization and the overlap of computations with other training tasks. Naturally, the exact statistics may vary depending on the specific hardware used. For example, the reduction in training speed when increasing the rank from $k=8$ to $k=64$ is mostly within 10%. Besides, in this setting, the training speed only shows a significant reduction when $k=256$. When using the same rank $k$ across models of different sizes,the number of layers must be considered, since layers are processed sequentially. For example, GPT-2 125M has 12 layers, while GPT-2 345M has 24 layers.
>
> This observation underscores the importance of our adaptive rank mechanism, which is designed to minimize computational time and memory usage by dynamically adjusting the rank as needed. As mentioned in Section 4.1, with the adaptive rank mechanism, Adapprox maintains a higher rank only during the initial training stages and transitions to a relatively low rank for the remainder of training. For instance, as shown in Table 2, the mean rank during training is approximately 1.4 for GPT-2 117M and 7.7 for BERT 345M. This will accelerate the overall training process.
>
> In addition, for the primary purpose of saving memory, the rank $k$ is typically not set too high. Otherwise, there may be little to no memory savings, defeating the goal of memory efficiency.

---

> ### Author Response · Authors · 2024-11-22
> **Part 2**
>
> **Response to Question 3**
>
> Thank you for your suggestion. Table 3 presents the zero-shot results of pre-trained GPT-2 models using various optimizers. The pre-training times for these optimizers are shown in Table 4, where Adapprox incurs a latency increase of approximately 6% compared to Adam on GPT-2 117M. Memory usage for Adapprox is detailed in Table 2 and depends on the selected rank $k$. For Adafactor and CAME, as they utilize a fixed rank-1 ($k=1$) approximation, their memory usage is nearly half that of Adam. We will clarify this information further in the revised manuscript.
>
> In summary, Adapprox typically requires more time than AdamW but uses less memory. While it requires more memory than Adafactor and CAME, it achieves better training performance. Overall, our method provides a more balanced trade-off between memory utilization and training efficiency.
>
> **Response to Question 4**
>
> Thank you for your suggestion. Below, we provide both a theoretical analysis and experimental results regarding memory utilization. We will incorporate this information into the revised manuscript accordingly.
>
> ***Theory Analysis of Memory Usage***
>
> In low-rank approximation for the second moment matrix, memory utilization is dominated by the storage of the decomposed matrices $Q$ and $U$. For a second-moment matrix $V$ of size $m\times n$, the memory usage under low-rank approximation is:
>
> $$
> \text{Memory Usage} = k \cdot (m+n),
> $$
>
> where $k$ is the rank of the approximation. In contrast, storing the full matrix $V$ requires $m\times n$ elements. The ratio of memory saved can be expressed as:
>
> $$
> \text{Memory Savings} = 1 - \frac{k \cdot (m+n)}{m\cdot n}.
> $$
>
> For larger models (with higher $m$ and $n$), the relative savings increase as long as $k$ remains small compared to $m$ and $n$. However, increasing $k$ leads to higher memory usage, and for sufficiently large $k$, the savings diminish. For example, consider a second moment matrix $V$ of $1024\times 1024 (m=n=1024)$:
>
> - with $k=16$, the memory savings are $1-16\times(1024+1024)/1024^2 = 96.88\%$;
> - with $k=256$, the memory savings are $1-256\times(1024+1024)/1024^2 = 50.00\%$.
>
> In addition, to achieve memory savings in this example, $k$ must be less than $512$.
>
> ***Experimental Results***
>
> To validate the efficiency, we measured the memory utilization of optimizer states for various ranks $k$ across GPT-2 models of different sizes during active training. The memory usage includes both the low-rank approximation and other optimizer states (e.g., first moment). Below are the results (in MB):
>
> |Model|k=1|k=8|k=16|k=32|k=64|k=128|k=256|AdamW|
> |-|-|-|-|-|-|-|-|-|
> |GPT-2 125M|475.93|481.27|487.39|499.61|524.06|572.95|670.74|949.40|
> |GPT-2 345M|1356.47|1368.40|1382.02|1409.28|1463.79|1572.81|1790.85|2707.09|
> |GPT-2 1.1B|3550.78|3573.88|3600.29|3653.09|3758.69|3969.90|4392.31|7089.92|
>
> As shown in the results above, when $k$ is small (e.g., 1 or 8), memory utilization for optimizer states is reduced by nearly half compared to AdamW, demonstrating significant savings due to compressed second moment. This reduction is particularly beneficial for larger models, such as GPT-2 1.1B.
>
> As $k$ increases, memory usage grows, approaching that of AdamW at higher ranks. For example, at $k=256$, memory utilization for GPT-2 1.1B is 38% lower than AdamW. This underscores the importance of selecting an appropriate $k$ to balance memory savings and approximation accuracy.
>
> These results highlight the efficiency of low-rank approximations in reducing memory utilization for optimizer states, particularly when $k$ is small. We will include these findings in the revised manuscript to further illustrate the memory-saving potential of our method.

---

### Official Review · Reviewer_hXQt · 2024-11-11

**Soundness:** 2
**Presentation:** 2
**Contribution:** 3
**Rating:** 5
**Confidence:** 4

**Summary:**

The paper presents Adapprox, a memory-efficient optimization algorithm that reduces memory usage through factorization of the second moment matrix in the Adam algorithm. The key distinction from previous approaches, such as Adafactor and CAME, lies in its adaptive rank selection capability during training, whereas earlier methods were limited to rank-1 approximations. To enable this dynamic rank adjustment during training, the authors incorporated a fast randomized SVD decomposition algorithm. The method's performance and computational efficiency are evaluated through experiments on pre-training and fine-tuning of GPT-2 and BERT models.

**Strengths:**

1. Given the current trends in model scaling, the chosen problem of memory-efficient optimization is highly relevant and represents an important research direction.

2. The Streamlined Randomized Subspace Iteration algorithm demonstrates good performance in terms of both computational efficiency and approximation quality, with potential applications across various domains.

3. The observation regarding singular values of the second moment matrix is particularly insightful, making the proposed higher-rank factorization algorithm both novel and valuable.

**Weaknesses:**

1. The quality of experimental sections, including setup, descriptions and results presentation (both in Sections 3.2 and 4), requires improvement. See specific concerns, questions and suggestions in the section Questions below.

2. The absence of experimental code is a notable limitation, particularly since one of the paper's main contributions is the efficient implementation of the Streamlined Randomized Subspace Iteration algorithm.

3. The comparison baselines could be expanded to include other recent memory-efficient optimization algorithms, such as those presented in [1] and [2].

[1] Zhao et al., 2024. GaLore: Memory-Efficient LLM Training by Gradient Low-Rank Projection. https://icml.cc/virtual/2024/poster/33390

[2] Zhang et al., 2024. Adam-mini: Use fewer learning rates to gain more. https://arxiv.org/abs/2406.16793

**Questions:**

Major concerns:

1. Section 3.2 requires clarification regarding the use of Adafactor in experimental comparisons. As Adafactor is basically an optimization algorithm [1] rather than a matrix decomposition method, its inclusion requires further explanation. Therefore, it's unclear what exactly is being compared in Figure 2:

* In Figure 2(a), what is meant by Adafactor "approximation"? How exactly is it computed?
* In Figure 2(b), what is Adafactor time? Is this the time of an Adafactor optimizer step? If so, comparing an optimization algorithm step with a matrix decomposition algorithm seems incorrect.
* Also, In Figure 2(b), it's impossible to discern the difference between Adafactor and S-RSI. I would advise the authors to change the presentation, for example, by providing a table instead.

2. Regarding Figure 4, several clarifications are needed:

* What sequence length was employed in the experiments? For language model training, the total batch size measured in tokens is a crucial parameter (see, for example, Table 2.1 in [2]).

* The hyperparameter selection process described in lines 393-399 requires elaboration. The term "uniform training parameters" needs clarification - does this indicate identical hyperparameters across all methods? Please also specify the methodology for "empirical testing": Were these parameters optimized for AdamW or Adapprox? Additionally, the hyperparameter search space should be detailed. This concern is particularly relevant for CAME, as the results (especially in Figure 4c) show initial superiority but deteriorating performance in later stages. This pattern often indicates a potentially sub-optimal (too large) learning rate choice.

* The simultaneous reporting of both perplexity and evaluation loss appears redundant, given that perplexity is simply the exponential of the evaluation loss.

* Figure 4's current visualization makes it challenging to distinguish between Adam, Adafactor, and Adapprox performance. I would suggest supplementing the figure with a table presenting final performance metrics for clearer comparison.

3. Regarding Table 3, the choice of zero-shot testing for non-instruct models raises concerns. Unlike instruct-tuned models that are designed for direct task completion, non-instruct models are highly sensitive to prompt engineering, making zero-shot evaluation potentially unreliable. I suggest to the authors:

* Justify why this evaluation setup is appropriate (e.g., by citing similar studies that use zero-shot accuracy for non-instruct models of comparable size)
* Provide either a detailed description of their zero-shot testing setup or reference the established protocol they followed, if any

4. Regarding Section 4.3 and Table 5:

* The experimental objective is unclear. If the purpose was to evaluate the optimizer's fine-tuning performance, why weren't all models initialized from the same starting point (e.g., the official BERT checkpoint)? The current setup, using different pre-trained results, creates uneven initial conditions and potentially biases the comparison.

* The comparison methodology appears incomplete, particularly the absence of PEFT methods. Given the paper's focus on memory efficiency, PEFT would serve as a natural and important baseline.

* The experimental details are insufficient - specifically, the hyperparameters used in these experiments should be explicitly stated.

Minor concerns:

1. Regarding Algorithm 3, line 347, where the square root of $V_t$ is computed: This operation implicitly assumes that all entries in $V_t$ are non-negative. However, since the $QU^{\top}$ decomposition used in line 344 is an approximation of truncated SVD rather than an exact representation, there's a concern about potential negative entries in this product $QU^{\top}$ (due to the approximation errors and floating-point precision errors) and hence in $V_t$. Could the authors please clarify how this edge case is handled or, preferably, provide a proof or explanation for why negative entries cannot occur in $V_t$ despite the approximate nature of the decomposition?

2. The paper would benefit from an additional ablation study on rank adaptation necessity. Given that the system already allocates memory for $k_{max}$ at the start of the training (line 377), maintaining this fixed rank throughout might potentially yield better performance, eliminating the need for dynamic adaptation.

3. Several typos and suggestions regarding writing:

* There's an inconsistency between Table 5 and Section 4.3: one mentions GPT 345M while the other refers to BERT 345M.

* The dimensions of matrix $U$ are inconsistent between line 161 and line 195. Adding explicit dimensions in Algorithm 1 also would be helpful.

* The units for Mean Computation Time in Figure 2(b) are not specified.

* In Algorithm 3's Inputs, $V_0$ is listed as an $m\times n$ matrix, suggesting this matrix $V$ is maintained in memory throughout training. It would be clearer to explicitly list matrices $Q$ and $U$ instead, as they are the only ones stored between optimization steps.

[1] Shazeer & Stern, 2018. Adafactor: Adaptive learning rates with sublinear memory cost. https://proceedings.mlr.press/v80/shazeer18a/shazeer18a.pdf

[2] Brown et al., 2020. Language Models are Few-Shot Learners. https://arxiv.org/abs/2005.14165

---

> ### Author Response · Authors · 2024-11-21
> **Part 1**
>
> **Response to Major Question 1**
>
> Thank you for your detailed observations and valuable feedback. We acknowledge that our presentation may have caused some confusion. Below, we provide clarifications and will ensure these points are addressed more clearly in the revised manuscript:
>
> - The term "Adafactor approximation" refers specifically to the low-rank matrix approximation method proposed and employed within the Adafactor algorithm (this method is also used in CAME). We compare this approximation with our S-RSI, focusing on approximation accuracy rather than the broader optimization process. Specifically, for a matrix $A \in R_{\geq 0}^{m \times n}$, Adafactor uses two vectors $R \in R_{\geq 0}^{m \times 1}$  and $S \in \mathbb{R}_{\geq 0}^{1 \times n}$, where $R = A1_n$, and $S = 1_m^\top A/1_m^\top A1_n$. Here, $R$ represents the row sum of $A$, and $S$ represents the normalized column sum of $A$. In Figure 2(a), we present the mean approximation error, calculated as $1/s\sum_i^s||A_i - R_i S_i||_F$, where $s$ denotes the number of samples.
> - The "Adafactor time" reflects the time required to compute the low-rank approximation within Adafactor's implementation. Specifically, as mentioned in the first point, this includes the time taken to compute $R$ and $S$. To ensure a fair comparison, we implemented this single module independently for evaluation.
> - To address this, we will enhance clarity by adding a table in our revised version to summarize the numerical results, making the comparisons easier to interpret. We acknowledge that S-RSI is slower than Adafactor's low-rank approximation method, which involves only summation and element-wise division. However, Figure 2(b) also highlights that S-RSI’s runtime on GPUs scales sub-linearly with rank $k$, contrary to theoretical expectations (as outlined in Table 1 of our manuscript).
>
> **Response to Major Question 2**
>
> Thank you for your insightful feedback regarding Figure 4. We appreciate your thorough observations and provide the following clarifications:
>
> - In our experiments, the sequence length was set to 1024 tokens, with a batch size of 128. Therefore, the total batch size (measured in tokens) is 131,072 tokens per iteration. We will clarify this in the revised manuscript.
> - There are three points:
>   - The term "uniform training parameters" refers to the use of identical hyperparameters (e.g., batch size, number of training iterations, number of warmup iterations, peak learning rate, minimum learning rate, learning rate scheduler, weight decay, etc.) across all methods to ensure a fair comparison.
>   - Regarding the "empirical testing" methodology, we determined hyperparameters (primarily the peak learning rate and minimum learning rate) through grid search, prioritizing stability and competitive performance for the baseline methods. The hyperparameter search space for learning rates was $[10^{-5}, 10^{-2}]$.
>   - For CAME, we also carefully considered its behavior when selecting the learning rate. As you pointed out, its convergence pattern often show rapid reduction in training loss during the initial stages but slower progress in later stages. We attribute this to CAME's inherent design. Specifically, in Algorithm 2 of CAME (lines highlighted in blue), a "confidence score" mechanism is employed to adjust the step size of updates, where the scale factor is $1/\sqrt{S_t}$ in their notation. In the experiments, we observed that $1/\sqrt{S_t}$ is almost always greater than 1, effectively enlarging the learning rate. This explains why CAME converges more quickly in the early stages but slows down in later stages. Moreover, this mechanism can sometimes cause instability in CAME's convergence. Our experimental results reflect these characteristics, which we believe highlight an inherent feature of CAME's design.
> - In the revised manuscript, we will simplify the presentation by reporting only perplexity, which is more interpretable for practitioners.
> - Thanks for your suggestion. To address this, we will include a supplementary table or a detailed illustration summarizing the final performance metrics within the context.

---

> ### Author Response · Authors · 2024-11-21
> **Part 2**
>
> **Response to Major Question 3**
>
> - We chose zero-shot testing for pre-trained models to ensure a consistent comparison across models without relying on task-specific fine-tuning or prompt engineering. This evaluation metric is inspired by a series of post-training compression studies, such as SparseGPT [1] and Wanda [2], which also employ zero-shot accuracy alongside perplexity to measure the performance of pre-trained models and their compressed variants.
> - We conducted zero-shot evaluation using the LM Harness library [3], which facilitates standardized testing of language models. The saved pre-trained models were loaded into this library, and the zero-shot evaluation mode was activated to generate the corresponding results.
>
> [1] Frantar, Elias, and Dan Alistarh. "Sparsegpt: Massive language models can be accurately pruned in one-shot." International Conference on Machine Learning. PMLR, 2023.
>
> [2] Sun, Mingjie, et al. "A Simple and Effective Pruning Approach for Large Language Models." The Twelfth International Conference on Learning Representations.
>
> [3] Leo Gao, Jonathan Tow, Stella Biderman, Sid Black, Anthony DiPofi, Charles Foster, Laurence Golding, Jeffrey Hsu, Kyle McDonell, Niklas Muennighoff, et al. A framework for few-shot language model evaluation. Version v0. 0.1. Sept, pp. 8, 2021.
>
> **Response to Major Question 4**
> - The objective of this experiment was to evaluate the optimizer's performance in a complete end-to-end training pipeline, encompassing both the pre-training and fine-tuning stages. To align with this goal, we performed fine-tuning on pre-trained models optimized with the respective optimizers.
> - We agree that PEFT methods are relevant baselines for a study focused on memory efficiency in fine-tuning, and we acknowledge the valuable contributions of methods like LoRA [1]. However, our primary focus was on comparing full-model optimization approaches leveraging low-rank matrix approximation. Therefore, we selected baselines such as Adafactor and CAME for this study.
> - The primary hyperparameter in this stage is learning rate, which can vary across optimizers and tasks. To ensure a fair comparison, we conducted all experiments on BERT-345M using learning rates from the set [1e-5, 5e-5,1e-4, 2e-4] respectively and retained the best results for each optimizer. Similarly, for GPT-2 117M, we tested learning rates from the set [2e-5, 4e-5, 5e-5], again keeping the best results for each optimizer. For batch size, we set 128. Other optimizer specific hyperparameters are set the same as pre-training stage as we mentioned in Section 4. The batch size was fixed at 128 for all experiments, and other optimizer-specific hyperparameters were set the same as in the pre-training stage, as described in Section 4. We will add these details to the revised manuscript to ensure clarity.
>
> [1] Hu, Edward J., et al. "LoRA: Low-Rank Adaptation of Large Language Models." International Conference on Learning Representations.
>
> **Response to Minor Question 1**
>
> Thank you for your insightful comment. We perform a projection step at this stage, which was inadvertently omitted from our pseudocode. This step clamps negative elements of $QU^\top$ to zero. This projection is necessary not only to ensure non-negativity but also to potentially reduce the approximation error as $V_t \geq 0$. Mathematically, there is:
>
> $$
> ||V_t - [Q_tU_t^\top]_+||_F \leq ||V_t - Q_tU_t^\top||_F
> $$
>
> where $[x]_+$ denotes the operation of clamping negative values to zero.
>
> We will clarify this procedural detail and its justification in the revised version of our manuscript.
>
> **Response to Minor Question 2**
>
> Thank you for your comment. Dynamic rank adaptation was introduced to balance memory efficiency and approximation accuracy. While memory for $k_0=k_{max}$ is allocated upfront, maintaining a fixed rank throughout training could lead to unnecessary computational overhead and memory usage, particularly in scenarios where a smaller rank suffices (e.g., during the later stages of training or for less complex structures). Dynamic adaptation allows the rank to adjust based on the training requirements, optimizing both resource utilization and performance. In contrast, a fixed-rank approach may allocate excessive resources uniformly, which might not translate into proportional gains in accuracy.

---

> ### Author Response · Authors · 2024-11-21
> **Part 3**
>
> **Response to Minor Question 3**
>
> Thank you for your detailed observations. We address your points as follows:
> - The correct model is BERT 345M, and we will revise Table 5 to ensure consistency with the rest of the manuscript.
> - Thank you for pointing out the inconsistency in the dimensions of matrix $U$. We intended $U$ to be $n \times k$, with $U^\top$ as $k \times n$. We will review and correct this discrepancy throughout the manuscript and explicitly include the dimensions of the matrices in Algorithm 1 to improve clarity.
> - The units are seconds, and we will update the figure caption to reflect this explicitly.
> - Thank you for this insightful suggestion. We will revise the inputs list of Algorithm 3.
>
> We greatly appreciate your detailed and valuable feedback and will incorporate these revisions to improve the clarity, accuracy, and overall readability of the manuscript.
>
> **Response to Weakness 1**
>
> Thank you for your detailed and valuable feedback. We have carefully addressed your concerns and suggestions in the responses above.
>
> **Response to Weakness 2**
>
> Thank you for highlighting this important point. We plan to release the code upon acceptance of the manuscript. In the meantime, the pseudocode provided in the manuscript can be used to reproduce our results.
>
> **Response to Weakness 3**
>
> Thank you for your suggestion to expand the comparison baselines. We have conducted additional experiments to include comparison results with GaLore [1]. Regarding Adam-mini, as it is also a submission to ICLR 2025, we plan to conduct a comprehensive comparison with it in future work. Below, we provide details of the additional results on GPT-2 125M.
>
> |**Method**|**Specific Parameters**|
> |-|-|
> |**Common**| Batch size = 8, Learning rate = 1e-4, Weight decay = 0.01, Steps=10K， Sequence length = 1024|
> |**SGD**|N/A|
> |**SGDM**|$\beta_1 = 0.9$|
> |**AdamW**|$\beta_1 = 0.9, \beta_2 = 0.999$|
> |**Adafactor**|$\beta_1 = 0.9, \beta_2 = 0.999$|
> |**CAME**|$\beta_1 = 0.9, \beta_2 = 0.999, \beta_3 = 0.9999$|
> |**GaLoreAdamW**|Rank = 128, Update proj gap = 200, Scale = 0.25, Proj type = std|
> |**Adapprox**|$k_0=k_{max}=k_{min}=1, l=1, p=5$|
>
> (*Those parameters not mentioned are set as default. All experiments were performed on a single NVIDIA RTX 4090 GPU.*)
>
> The training loss recorded at every 1,000 steps and the overall training time for each method are presented in the table below:
> |**Method**|Step 1K|Step 2K|Step 3K|Step 4K|Step 5K|Step 6K|Step 7K|Step 8K|Step 9K|Step 10K|Time (min)|
> |-|-|-|-|-|-|-|-|-|-|-|-|
> |**SGD**|8.49|8.09|8.31|8.04|7.96|7.82|8.02|7.67|7.60|7.12|**22.49**|
> |**SGDM**|7.63|7.23|7.37|7.12|7.05|6.92|7.10|6.78|6.73|6.28|**22.71**|
> |**AdamW**|5.66|5.16|5.05|4.81|4.69|4.50|4.64|4.34|4.23|3.99|**23.81**|
> |**Adafactor**|5.64|5.15|5.04|4.80|4.69|4.50|4.64|4.35|4.24|4.00|**26.76**|
> |**CAME**|5.42|4.91|4.81|4.57|4.48|4.30|4.45|4.18|4.07|3.86|**29.14**|
> |**GaLoreAdamW (Full rank, Actually adamW)**|5.67|5.18|5.07|4.82|4.70|4.51|4.65|4.36|4.24|4.01|**23.20**|
> |**GaLoreAdamW (rank=128)**|5.44|5.61|5.60|5.14|5.04|5.08|4.82|4.85|5.02|5.17|**24.97**|
> |**Adapprox (Ours)**|5.63|5.13|4.99|4.74|4.62|4.41|4.52|4.20|4.06|3.81|**28.84**|
>
> (*In the initial submission of this rebuttal, GaLoreAdamW's rank decomposition mechanism was not enabled, and the results corresponded to AdamW. We have now corrected this oversight and included the accurate data for rank = 128.*)
>
> The results above align with the conclusions presented in our manuscript. While SGD and SGDM require the least memory and computational cost, they converge significantly slower. This reinforces our belief that incorporating the second moment is essential for effectively training LLMs. CAME may reduce the training loss quickly in the initial stages but slows down as training progresses. and Adafactor demonstrates convergence behavior similar to that of AdamW. As GaLoreAdamW compresses the gradient, which is crucial for guiding the optimization direction, its performance drops a lot and becomes unstable. Our Adapprox method demonstrates stable performance and achieves the lowest final training loss, even when the rank for approximation is aggressively fixed at a constant value of 1. This also indicates that, like Adafactor and CAME, we compress the second moment matrix using two vectors, effectively eliminating nearly all memory usage associated with the second moment. However, we acknowledge that Adapprox introduces additional time overhead, a common trade-off for memory-efficient optimizers that involve extra computation.
>
> [1] Zhao et al., 2024. GaLore: Memory-Efficient LLM Training by Gradient Low-Rank Projection.

---

> > ### Comment · Reviewer_hXQt · 2024-11-24
> > **Follow-up questions part 1**
> >
> > I would like to thank the authors for their detailed response. Some of my concerns have been adequately addressed. However, I still have several follow-up questions.
> >
> > **Regarding Major Question 2.**
> >
> > >use of identical hyperparameters (e.g., batch size, number of training iterations, number of warmup iterations, peak learning rate, minimum learning rate, learning rate scheduler, weight decay, etc.) across all methods to ensure a fair comparison.
> >
> > I respectfully disagree with the authors' assertion that identical hyperparameters, especially identical learning rates, ensure fair comparison. It is well-established that optimal learning rates for SGD and Adam differ significantly (primarily because the update of Adam-like algorithms approximately equals $\text{lr} \cdot \text{sign}(g)$). Moreover, even among Adam-like algorithms (with sign-like updates), the optimal learning rate can vary considerably (as demonstrated in [1,2]).
> >
> > This concern particularly applies to the CAME results. While the authors' explanation that the CAME algorithm is generally more aggressive in early training stages might be valid, an alternative explanation that I proposed in my initial review remains possible - specifically, that the chosen learning rate (shared for all the optimizers) might be too high for CAME. If the authors had separately tuned the learning rate for each algorithm, it would have been clearer which hypothesis is correct.
> >
> > To sum up, while I acknowledge that conducting extensive hyperparameter searches for parameters such as $\beta_1, \beta_2$, weight_decay, and others might be computationally prohibitive, I maintain that learning rates should be tuned separately for different methods. This approach is, in fact, standard practice in optimization literature, as evidenced in [3,4], among others.
> >
> > >prioritizing stability and competitive performance for the baseline methods.
> >
> > I find that this explanation does not constitute a formal selection methodology and does not fully address my concern. The authors' baseline comparison included three methods. While it is theoretically possible that one set of hyperparameters could be optimal for all three methods simultaneously, such a situation is not guaranteed in the general case. Therefore, it remains unclear how the selection between the two hyperparameter sets was made in cases where, for instance, one set performed better for AdamW while the other set was superior for Adafactor.
> >
> > **Regarding Major Question 4.**
> >
> > >The objective of this experiment was to evaluate the optimizer's performance in a complete end-to-end training pipeline, encompassing both the pre-training and fine-tuning stages. To align with this goal, we performed fine-tuning on pre-trained models optimized with the respective optimizers.
> >
> > To the best of my knowledge, this practice is not a standard pipeline for evaluating optimizers, particularly because pre-training and fine-tuning stages typically require different optimal hyperparameters. Therefore, for a comprehensive comparison of the full pipeline, optimal hyperparameters should be tuned separately for both stages. I would recommend that the authors provide references to papers with similar experimental setups to justify using this approach for optimizer comparison.
> >
> > >Other optimizer specific hyperparameters are set the same as pre-training stage as we mentioned in Section 4. The batch size was fixed at 128 for all experiments, and other optimizer-specific hyperparameters were set the same as in the pre-training stage, as described in Section 4.
> >
> > I would recommend that the authors provide a complete specification of all hyperparameters, including: 1) number of epochs, 2) warmup ratio, 3) sequence length, 4) scheduler configuration, and other relevant parameters
> >
> > This is particularly important as these parameters typically vary across different fine-tuning datasets (see, e.g. [5]), and their explicit documentation would enhance the reproducibility of the experiments.

---

> ### Comment · Reviewer_hXQt · 2024-11-24
> **Follow-up questions part 2**
>
> **Regarding Response to Weakness 3**
>
> I have several concerns about these tables:
>
> * As previously mentioned, different methods require different optimal learning rates. This concern is particularly relevant for SGD and GaLore, where the original paper [4] used a learning rate of 1e-2 (100 times larger) for a model of comparable size and similar number of steps.
>
> * Generally, the learning rate selection appears questionable. For models of this scale, 1e-4 is notably too small (see, e.g. [6], Table 2.1).
>
> * Other hyperparameter choices also raise concerns. The batch size, in particular, is extremely small (see [6], Table 2.1). It's also known that an inadequately small batch size can severely impact transformer performance (see, e.g. [7] Figure 5, 6).
>
> * With batch size = 8, sequence length 1024, and 10k steps, the total number of tokens processed is approximately 82M, which is substantially below even the Chinchilla-optimal token count (which should be ~20x the parameter count) [8]. This means the authors are only comparing performance during the very early stages of model training.
>
> * The scheduler type and number of warmup steps used in the experiment are not specified. Proper warmup is known to be crucial for training transformers (see, e.g., [9] Table 1 and [7] Figure 8).
>
> To summarize, I believe this experimental setup does not provide a sufficient basis for comparing optimizer performance. For a fair comparison of optimization algorithms, I would recommend that the authors conduct experiments with GaLore using the same setup as for other baselines in original paper.
>
> [1] Chen, et al. "Symbolic discovery of optimization algorithms.", NeurIPS, 2023.
>
> [2] Pagliardini et al. "The ademamix optimizer: Better, faster, older.", arXiv:2409.03137, 2024.
>
> [3] Zhang et al. "Adam-mini: Use fewer learning rates to gain more.", arXiv:2406.16793, 2024.
>
> [4] Zhao et al. "GaLore: Memory-Efficient LLM Training by Gradient Low-Rank Projection.", ICML, 2024.
>
> [5] Hu et al. "LoRA: Low-Rank Adaptation of Large Language Models.", ICLR, 2022.
>
> [6] Brown et al. "Language models are few-shot learners." arXiv:2005.14165, 2020.
>
> [7] Popel et al. "Training tips for the transformer model." arXiv:1804.00247, 2018.
>
> [8] Hoffmann et al. "Training compute-optimal large language models." arXiv:2203.15556, 2022.
>
> [9] Shazeer & Stern. "Adafactor: Adaptive learning rates with sublinear memory cost." ICML, 2018.

---

> > ### Author Response · Authors · 2024-11-29
> > **Response to Follow-up Questions Part 1**
> >
> > **Regarding to Follow-up of Question 2 and Weakness 3**
> >
> > We acknowledge your point that optimal learning rates (and potentially other hyperparameters) often vary across optimization methods. Our decision to use identical hyperparameters was guided by the following rationale:
> >
> > - Given the computational cost of tuning hyperparameters for each optimizer, we prioritized stability and ensured that the optimizers were evaluated under controlled, consistent conditions. This approach allowed for a clearer comparison of their relative performance. Specifically, for the latest baseline in compressing the second moment via low-rank approximation (CAME) [1], we conducted additional experiments with varying learning rates, as detailed later, to optimize its performance within our specific setting.
> >
> > Regarding learning rate selection, we tested learning rates within the set {1e-5, 5e-5, 1e-4, 2e-4} and found 1e-4 to be a generally effective choice. Smaller learning rates caused slower convergence, while larger ones led to instability or divergence, ultimately slowing convergence.
> >
> > For SGD, we noted that Adam-like optimizers generally outperform SGD without momentum on language tasks, as shown in prior work [2, 3]. This highlights the necessity of second-moment estimation and motivates efforts to compress it rather than eliminate it to save memory. To further compare SGD with other methods, we included results with larger learning rates. Similarly, we tested GaLore across various learning rates to provide a more comprehensive evaluation under our experimental setting.
> >
> > We acknowledge that the batch size and number of steps used in our experiments are smaller than those typically seen in practical applications, due to the limitations of our available hardware and the need to meet constrained response times. However, we believe that these demo examples effectively capture the performance trends of the various optimizers. This approach is also employed in GaLore for quick comparisons, as demonstrated in Figure 3 of GaLore's paper [4], where LLaMA 1B is pre-trained on the C4 dataset for 10K steps.
> >
> > We detailed the setting of our experiments (including details we did not show previously):
> >
> > |**Method**|**Specific Parameters**|
> > |-|-|
> > |**Common**| Batch size = 8, Weight decay = 0.01, Steps=10K, Sequence length=1024, Warm up steps= 0.05 * 10K = 500, Scheduler type: linear (warm up to learning rate and then decay to zero linearly)|
> > |**SGD**|N/A|
> > |**SGDM**|$\beta_1 = 0.9$|
> > |**AdamW**|$\beta_1 = 0.9, \beta_2 = 0.999$|
> > |**Adafactor**|$\beta_1 = 0.9, \beta_2 = 0.999$|
> > |**CAME**|$\beta_1 = 0.9, \beta_2 = 0.999, \beta_3 = 0.9999$|
> > |**GaLoreAdamW**|Rank = 128, Update proj gap = 200, Scale = 0.25, Proj type = std|
> > |**Adapprox**|$k_0=k_{max}=k_{min}=1, l=1, p=5$|
> >
> > (*Those parameters not mentioned are set as default.*)
> >
> > We summarized the previous and new results below:
> >
> > |**Method**|Step 1K|Step 2K|Step 3K|Step 4K|Step 5K|Step 6K|Step 7K|Step 8K|Step 9K|Step 10K|
> > |-|-|-|-|-|-|-|-|-|-|-|
> > |**SGD (lr=1e-4)**|8.49|8.09|8.31|8.04|7.96|7.82|8.02|7.67|7.60|7.12|
> > |**SGD (lr=1e-2)**|6.82|6.44|6.54|6.29|6.25|6.12|6.29|6.15|5.93|5.54|
> > |**SGD (lr=2e-1)**|6.74|6.34|6.22|5.90|5.83|5.69|5.84|5.54|5.47|5.12|
> > |**SGD (lr=5e-1)**|diverged|
> > |**SGDM (lr=1e-4)**|7.63|7.23|7.37|7.12|7.05|6.92|7.10|6.78|6.73|6.28|
> > |**SGDM (lr=1e-2)**|5.67|5.65|5.75|5.69|5.68|5.50|5.42|5.39|5.23|5.03|
> > |**SGDM (lr=1e-1)**|diverged|
> > |**AdamW (lr=1e-4)**|5.66|5.16|5.05|4.81|4.69|4.50|4.64|4.34|4.23|3.99|
> > |**Adafactor (lr=1e-4)**|5.64|5.15|5.04|4.80|4.69|4.50|4.64|4.35|4.24|4.00|
> > |**CAME (lr=1e-5)**|5.77|5.40|5.39|5.21|5.16|5.02|5.20|4.92|4.84|4.57|
> > |**CAME (lr=5e-5)**|5.44|4.96|4.87|4.67|4.59|4.42|4.59|4.31|4.22|3.99|
> > |**CAME (lr=1e-4)**|5.42|4.91|4.81|4.57|4.48|4.30|4.45|4.18|4.07|3.86|
> > |**CAME (lr=2e-4)**|5.47|5.04|4.98|4.76|4.68|4.49|4.65|4.37|4.26|4.03|
> > |**CAME (lr=3e-4)**|5.55|5.25|5.36|5.17|5.14|5.02|5.18|4.92|4.38|4.54|
> > |**GaLoreAdamW (lr=1e-4)**|5.44|5.61|5.60|5.14|5.04|5.08|4.82|4.85|5.02|5.17|
> > |**GaLoreAdamW (lr=1e-3)**|5.41|4.97|4.86|4.61|4.50|4.41|4.44|4.16|4.03|3.80|
> > |**GaLoreAdamW (lr=1e-2)**|5.42|5.06|5.20|5.22|5.23|5.21|5.40|5.07|4.91|4.58|
> > |**GaLoreAdamW (lr=1e-1)**|diverged|
> > |**Adapprox (Ours)**|5.63|5.13|4.99|4.74|4.62|4.41|4.52|4.20|4.06|3.81|

---

> > ### Author Response · Authors · 2024-11-29
> > **Response to Follow-up Questions Part 2**
> >
> > Based on the above results, we draw the following conclusions:
> >
> > - Following your suggestion, we increased the learning rate for SGD/SGDM and observed that a higher learning rate accelerates convergence. However, even with a larger learning rate, these methods significantly underperform compared to Adam-like optimizers. Besides, too large learning rate may cause divergence. Additionally, SGDM tends to converge faster than SGD, highlighting the efficiency of incorporating both first and second moment.
> > - We evaluated CAME's performance with learning rates of 1e-5, 5e-5, 1e-4, and 2e-4. We found that smaller learning rates do not improve CAME's performance but instead slow down the convergence. Additionally, increasing the learning rate to 3e-4 negatively impacts CAME's performance.  As CAME represents a state-of-the-art approach for approximating the second moment via low rank approximation, we thought that it is reasonable to prioritize stability and ensure competitive performance against baseline methods appears reasonable for our previous comparative experiments.
> > - Following your suggestion, we increased the learning rate for GaLoreAdamW and observed that a higher learning rate than 1e-4 can accelerate convergence. However, excessively large learning rates such as $1 \times 10^{-1}$ lead to divergence. Additionally, its best performance within our search range appears at $\text{lr}=1 \times 10^{-3}$ with the final training loss 3.80.
> > - Additionally, we can further tune the parameter for Adapprox to explore its full potential and ensure fairness. Below, we present cases where larger learning rates are applied to Adapprox. Naturally, with larger learning rates, we may also adjust $k$ to enhance stability, demonstrating the adaptability and dynamic nature of our method. We set $k_0=k_{max}=8, k_{min}=2, l=5, p=5, lr=4\times 10^{-4}$ in the following case:
> >
> > |**Method**|Step 1K|Step 2K|Step 3K|Step 4K|Step 5K|Step 6K|Step 7K|Step 8K|Step 9K|Step 10K|
> > |-|-|-|-|-|-|-|-|-|-|-|
> > |Adapprox|5.55|5.10|4.96|4.69|4.58|4.35|4.46|4.15|4.01|3.76|
> >
> > Based on the results above, Adapprox is able to achieve the best performance in our test cases through parameter tuning.
> >
> > In summary, we agree that different optimizers have distinct optimal parameter settings, and an optimizer may not perform equally well in all scenarios. This variability is largely due to the highly non-convex nature of LLM optimization. To make comparisons more straightforward, we adopt a strategy that prioritizes stability, ensuring competitive performance against baseline methods in a given setting. From there, we tune (or in some cases, avoid tuning) the parameters of the candidate to evaluate its relative performance.
> >
> >
> > [1] Luo, Yang, et al. "CAME: Confidence-guided Adaptive Memory Efficient Optimization." Proceedings of the 61st Annual Meeting of the Association for Computational Linguistics (Volume 1: Long Papers). 2023.
> >
> > [2] Shazeer, Noam, and Mitchell Stern. "Adafactor: Adaptive learning rates with sublinear memory cost." International Conference on Machine Learning. PMLR, 2018.
> >
> > [3] Kunstner, Frederik, et al. "Noise is not the main factor behind the gap between sgd and adam on transformers, but sign descent might be." arXiv preprint arXiv:2304.13960 (2023).
> >
> > [4] Zhao, Jiawei, et al. "GaLore: Memory-Efficient LLM Training by Gradient Low-Rank Projection." Forty-first International Conference on Machine Learning.
> >
> > **Regarding to Follow-up of Major Question 4**
> >
> > Thank you for your valuable feedback. We appreciate your concern regarding the evaluation of optimizers across both pre-training and fine-tuning stages.
> >
> > We fully acknowledge the distinction between the hyperparameters required for pre-training and fine-tuning, as you mentioned (learning rates often vary across downstream tasks and optimizers). In response, we have included the relevant hyperparameter details in lines 401–406 of our revised manuscript.
> >
> > For downstream tasks, we individually adjust the learning rates within the following ranges: ${2 \times 10^{-5}, 4 \times 10^{-5}, 5 \times 10^{-5}}$ for GPT-2 117M and ${1 \times 10^{-5}, 5 \times 10^{-5}, 1 \times 10^{-4}, 2 \times 10^{-4}}$ for BERT 345M (*compared to pretraining, the relatively shorter fine-tuning cost allows for this more granular adjustment*). We then fine-tune the models that were pretrained with each evaluated optimizer, selecting the best results achieved under these learning rates. Additionally, we conduct 3 fine-tuning epochs for the GPT-2 117M model and 1 epoch for the BERT 345M model. The batch size is set to 128, the sequence length is 1024, the learning rate scheduler follows a cosine decay, and the warmup ratio is set to 0.01.
> >
> > Thank you once again for your insightful feedback! We hope this clarification addresses your concerns.

---

> > > ### Comment · Reviewer_hXQt · 2024-12-02
> > >
> > > I thank the authors for their detailed response. While the authors genuinely tried to improve their paper, some of my concerns remain unaddressed:
> > >
> > > * I still find the setup of additional experiments with GaLore [1] unconvincing. Issues with batch size and number of tokens remain unresolved, and I do not find the argument with the GaLore setup sufficiently convincing because their experiment in Figure 3 in [1] is, in my opinion, not indicative. The grid search for learning rates of AdamW and Adafactor was not conducted. I still suggest that the authors conduct experiments with GaLore (and Adam-mini [2]) in a full-fledged setup from Section 4.2 in the future.
> > >
> > > * The authors' explanation regarding the choice of configurations for the experiments in Section 4.2 is still not formal enough, and I do not rule out that the ranking of methods could change with a more thorough hyperparameter tuning.
> > >
> > > * The pipeline for evaluating optimizers in an end-to-end pipeline of both pre-training and fine-tuning still seems quite questionable.
> > >
> > > * In my understanding, the hyperparameters used for fine-tuning do not align with commonly accepted practices, especially the use of just one epoch for BERT (see, e.g., [3] Tables 9, 10 and [4] Section 3.2).
> > >
> > > Summarizing all the arguments I have laid out in the discussion, I believe the experimental setups chosen by the authors do not meet the necessary standards to draw conclusions about the quality of the considered optimizers. Therefore, I have decided to maintain my score.
> > >
> > > [1] Zhao et al. "GaLore: Memory-Efficient LLM Training by Gradient Low-Rank Projection.", ICML, 2024.
> > >
> > > [2] Zhang et al. "Adam-mini: Use fewer learning rates to gain more.", arXiv:2406.16793, 2024.
> > >
> > > [3] Hu et al. "LoRA: Low-Rank Adaptation of Large Language Models.", ICLR, 2022.
> > >
> > > [4] Houlsby et al. Parameter-Efficient Transfer Learning for NLP. arXiv:1902.00751, 2019.

---

> > > > ### Author Response · Authors · 2024-12-03
> > > >
> > > > Thank you for your comments and for taking the time to review our manuscript. We appreciate your detailed feedback and would like to provide further clarifications and address the concerns you raised.
> > > >
> > > > **GaLore Experiments Setup and Concerns**
> > > >
> > > > We understand your concerns regarding the setup of additional experiments with GaLore during the rebuttal period, particularly with respect to batch size and the number of tokens. As noted, our current configuration was influenced by hardware and time constraints. Despite these limitations, we demonstrated the performance of Adapprox under the chosen settings. We agree that exploring larger batch sizes and token configurations is necessary for a more thorough analysis, and we are committed to conducting these experiments in future work.
> > > >
> > > > Regarding Figure 3 of GaLore [1], the caption states: *"Applying GaLore to different optimizers for pre-training LLaMA 1B on the C4 dataset for 10K steps."* Additionally, as described in Section C.1 of the GaLore paper: *"We use a max sequence length of 256 for all models, with a batch size of 131K tokens."* This results in $10K \times 131K = 1.31B$ tokens, which is significantly less than 20 times the parameter count, as mentioned. However, we believe this figure effectively illustrates the convergence trends when GaLore is integrated with other methods. Similarly, our experiments during the rebuttal period demonstrated comparable convergence trends.
> > > >
> > > > While GaLore is designed to compress gradients and Adapprox focuses on compressing the second moment, we find the idea of integrating GaLore's framework with S-RSI and adaptive rank mechanism (or AS-RSI) in our manuscript. To the best of our knowledge, GaLore utilizes traditional SVD for low-rank approximation with a fixed rank. This integration could potentially improve the efficiency of low-rank matrix approximations. This direction is a promising direction for future exploration.
> > > >
> > > > **End-to-End Pipeline Evaluation**
> > > >
> > > > In our manuscript, we employed an end-to-end pipeline where models were pretrained from scratch using each optimizer and subsequently fine-tuned to compare their performance. This approach aims to simulate practical scenarios and reflect typical workflows in real-world applications. By evaluating both the pretraining and fine-tuning phases, we aim to provide a comprehensive assessment of the optimizers' capabilities. Additionally, we report the performance of these two stages separately in our manuscript to facilitate comparisons of the pretraining phase alone as well as the combined performance of both phases.
> > > >
> > > > We also want to emphasize that the primary goal of Adapprox is to save memory during full-parameter pretraining. While parameter-efficient fine-tuning methods (PEFT), such as LoRA [2], are highly effective at optimizing resource usage during fine-tuning, they are not applicable to pretraining. Adapprox addresses the distinct challenge of memory efficiency throughout the pretraining phase, complementing rather than competing with PEFT approaches.
> > > >
> > > > **Hyperparameters in Fine-tuning Stage**
> > > >
> > > > We appreciate your feedback regarding the hyperparameters used during fine-tuning. We observed that learning rates have a greater impact on optimizer performance during fine-tuning compared to pretraining. To address this, we conducted experiments with various learning rates and selected the best result for each optimizer to ensure a fair comparison. For other hyperparameters, such as batch size and number of epochs, we maintained uniform settings across all optimizers to ensure consistency and fairness. This approach was designed to facilitate an unbiased evaluation of the relative capabilities of each optimizer. To balance efficiency and resource constraints, we opted for a single epoch when fine-tuning BERT 345M. The results provide a relative comparison of the performance of each optimizer.
> > > >
> > > > As for the discussion of fine-tuning stage's configuration in Section 4.2, we will further polish it to enhance clarity and readability.
> > > >
> > > > ***Thank you again for your time and constructive suggestions.***
> > > >
> > > > [1] Zhao, Jiawei, et al. "GaLore: Memory-Efficient LLM Training by Gradient Low-Rank Projection." Forty-first International Conference on Machine Learning.
> > > >
> > > > [2] Hu, Edward J., et al. "LoRA: Low-Rank Adaptation of Large Language Models." International Conference on Learning Representations.

---

### Meta-Review · Area_Chair_6QXi · 2024-12-15

**Metareview:**

This paper introduces Adapprox, a memory-efficient optimization algorithm that improves upon the Adam optimizer by dynamically adjusting the rank of the second-moment matrix during training.

While the method and its motivation are sound, the experimental setup raises significant concerns. In particular, I agree with Reviewer hXQt regarding Weaknesses 2 and 3, which were not fully addressed during the rebuttal period. Additionally, I suggest that the authors provide a more detailed explanation of their choice of hyperparameters (e.g., learning rates).

Therefore, I recommend the paper for rejection.

**Additional Comments On Reviewer Discussion:**

Given mixed reviews, I relied on the experience and detailed evaluation provided by Reviewer hXQt, which I fully support after going through the reviews, discussion, and the paper itself.

---

### Decision · Program_Chairs · 2025-01-22

Reject